# Visual modulation of vestibular-evoked balance response disturbed by posterior cortical atrophy

Dilek Ocal[1] , Brian L. Day[2], Amy Peters[2], Matt Bancroft[2] , David Cash[1] , Diego Kaski[2], Natalie S. Ryan[1,3], Sebastian J. Crutch[1] and Keir X. X. Yong[1]

[1] *Dementia Research Centre, Department of Neurodegenerative Disease, UCL Queen Square Institute of Neurology, University College London, London, UK*

[2] *Department of Clinical and Movement Neurosciences, UCL Queen Square Institute of Neurology, London, UK*

[3] *UK Dementia Research Institute, UCL Queen Square Institute of Neurology, University College London, London, UK*

Handling Editors: Vaughan Macefield & Luke Henderson

The peer review history is available in the Supporting Information section of this article (https://doi.org/10.1113/JP288693#support-information-section).

**Abstract figure legend** Summary of key findings. (A) Effect of vision on balance responses to vestibular stimulation in controls. Mean galvanic vestibular stimulation (GVS)-evoked response time-course is shown without and with vision, illustrating how visual input 'dampens' the balance response. Direction conventions are indicated, with 0° oriented towards the anodal ear. (B) Differing effects of vision on early (<400 ms) post-stimulus responses. Compared to controls and typical Alzheimer's disease (Typical AD), participants with posterior cortical atrophy (PCA) show a reduced modulatory effect of vision on the initial response component, suggesting long-term feedforward decreases in visual gain. (C) Neuroanatomical correlates of abnormal balance responses in PCA. Exaggerated responses with vision were

Dilek Ocal completed a BSc and MSc in Neuroscience at King's College London, followed by a PhD at the Dementia Research Centre, University College London, investigating multisensory disturbances in Alzheimer's disease. Now a postdoctoral researcher at the same centre, her work integrates behavioural, cognitive and neuroimaging methods to examine brain–behaviour relationships. She is particularly interested in how these relationships contribute to the diverse behavioural and cognitive presentations within dementia syndromes. **Brian Day** is Emeritus Professor of Motor Neuroscience at University College London. His research has explored human sensorimotor function in health and neurological disease. He has been at the forefront in the development and application of non-invasive techniques for studying CNS function and circuitry in intact human studies, most notably transcranial magnetic and electric stimulation, and more recently vestibular stimulation. His current research is devoted to understanding multi-sensory processes that control and integrate whole-body action.

D. Ocal and B. L. Day contributed equally to this work.

The Journal of Physiology

associated with reduced grey matter volume in occipital and thalamic regions and with higher fractional anisotropy in occipito-thalamic projection fibres, implicating disrupted visual–vestibular integration pathways underlying altered balance control in PCA.

**Abstract** We have recently shown that perception of uprightness is disturbed in typical Alzheimer's disease (tAD) and posterior cortical atrophy (PCA; 'visual-variant Alzheimer's'); disturbances were attributed to disrupted spatial transformation of graviceptive information. Here, we extend investigations to the vestibular control of uprightness during stance using galvanic vestibular stimulation (GVS) under various proprioceptive and visual conditions. An appropriately directed response to GVS requires spatial transformation of the vestibular signal based on head orientation relative to the feet, while non-vestibular sensory information can modulate the magnitude of response. Balance responses were repeatedly evoked in healthy participants ($n = 21$) and participants with AD (tAD: $n = 18$; PCA: $n = 18$) under conditions evaluating proprioceptive–vestibular integration (head directed right, straight or left without vision), or visuo-vestibular integration (head directed straight with or without vision). Across head directions without vision, GVS-evoked response direction and magnitude were comparable across groups. These comparable responses without vision indicate sparing of vestibulo-motor systems and proprioceptive–vestibular integration predicated on transformations between head, body and leg coordinates. However, the modulating effect of vision on GVS-evoked response magnitude was decreased in PCA relative to control and tAD groups. This relatively decreased effect of vision in PCA was evident through the earliest mechanical indicator of response, consistent with reduced feed-forward effects of vision on vestibularly driven balance response. Exaggerated response magnitude with vision was associated with occipito-thalamic volumetric and projection fibre abnormalities in PCA. The findings suggest candidate pathways for visual modulation of vestibular balance control and dissociations between systems responsible for perceiving uprightness *versus* rapid sensorimotor balance control.

(Received 5 February 2025; accepted after revision 18 September 2025; first published online 17 October 2025)

**Corresponding author** Keir X. X. Yong: Dementia Research Centre, Department of Neurodegenerative Disease, UCL Queen Square Institute of Neurology, University College London, London, UK. Email: keir.yong@ucl.ac.uk

## Key points

- Accurate balance control relies on the integration of vestibular, proprioceptive and visual inputs, transformed across spatial reference frames.
- We examined whole-body balance responses to galvanic vestibular stimulation (GVS) under varying visual and proprioceptive conditions in people with posterior cortical atrophy (PCA), typical Alzheimer's disease (tAD) and controls.
- Without vision, GVS-evoked balance responses were comparable across groups with the head turned left, right or straight ahead, indicating spared proprioceptive–vestibular integration in patient groups.
- With vision, visual input had a reduced modulatory effect on GVS responses in PCA relative to tAD and control groups. This was evident at the earliest stage of mechanical response, suggesting reduced feedforward visual influences on balance control in PCA.
- Our findings implicate disrupted occipito-thalamic pathways in PCA-related balance disturbances. The findings reveal a dissociation between perceived uprightness and sensori-motor balance control, with implications for understanding disturbed spatial orientation and balance in dementia.

## Introduction

Stabilizing the body in space requires multiple sources of self-motion cues: visual signals from optic flow, vestibular signals of head acceleration, and somatosensory signals of joint position and contact pressure (Day & Fitzpatrick, 2005a). Accurately estimating the body's motion relative to the environment thus necessitates the integration of noisy and occasionally conflicting sensory signals encoded in different egocentric reference frames (e.g. head-, eye-, and body-centred; Dakin & Rosenberg, 2018). The integration of these multisensory inputs is mediated by multiple brain regions, particularly within the posterior cortical areas (Buneo et al., 2002; Duhamel et al., 1998). The posterior parietal cortex and occipito-temporal regions play crucial roles in processing visual information, multisensory integration and spatial transformations essential for constructing a coherent perception of body orientation and movement in space, and thus possibly for balance control (Buneo et al., 2002; Duhamel et al., 1998). Consequently, disorders that affect parietal regions and associated functions may compromise the ability to stabilize the body while standing.

Parietal vulnerability in Alzheimer's disease (AD) may contribute to the gait disorders and increased risk of falls reported in AD (Allan et al., 2009; Coughlan et al., 2018). This is particularly true for individuals living with posterior cortical atrophy (PCA), characteristically an atypical variant of AD featuring progressive decline in visual processing and non-visual posterior cortical functions despite relative sparing of memory, language and executive functions (Crutch et al., 2017; Tang-Wai et al., 2004). Reports suggestive of spatial disorientation and subjective balance disorders include PCA patients asking, 'Am I the right way up?' while sitting; commenting, 'I felt like I was about to fall off the edge of the world' when walking; and noting sensations of 'floating along the ceiling' after descending stairs (Crutch et al., 2018). Additionally, patients leaning to one side during walking or standing have been observed in PCA and late-stage AD (Crutch et al., 2018; van Engelen et al., 2020), consistent with disorientation in the vertical plane. While visual processing deficits due to occipital lobe degeneration are well-established in PCA (also known as 'visual dementia'), the impact of posterior parietal and occipital atrophy on sensorimotor mechanisms known to contribute to balance control in standing humans remains largely unclear.

We have previously shown that the *perception* of verticality, whether judged visually or haptically, is disturbed in AD (Day et al., 2022). Greater disturbances in verticality perception in PCA relative to typical AD (tAD) were attributed to disruption of a system responsible for spatially transforming graviceptive signals between body parts owing to posterior parietal damage. While superior temporo-parietal and thalamic regions have been implicated as key sites in verticality perception (Perennou et al., 2008), it is possible that subcortical structures play a dominant, or even exclusive, role in balance control. By targeting specific sensorimotor processes, we may be able to establish whether the cortical structures compromised in AD, which give rise to perceptual problems, also play a role in the control of human balance. Furthermore, we consider current findings in light of disturbances in verticality perception to better understand well-documented dissociations between perceptual and sensorimotor functions relevant to vestibular control of balance (Dalton et al., 2017; Wardman et al., 2003).

Here we report the results from one series of experiments in which we employed a discrete vestibular perturbation that is known to evoke a stereotyped balance response in healthy standing people. The vestibular perturbation consists of a small electric current delivered to the mastoid processes (galvanic vestibular stimulation; GVS) that acts to modulate the firing rate of semicircular canal and otolith afferents (Goldberg et al., 1984; Kwan et al., 2019). According to one model, the net effect of this vestibular afferent activity mimics a vestibular signal consistent with an apparent rotation of the head approximately about its roll axis (Fitzpatrick & Day, 2004). This modelled response has received experimental support in human studies involving rotation perception (Day & Fitzpatrick, 2005b), gait (Fitzpatrick et al., 2006) and quiet standing (Mian et al., 2010). When standing still with head upright, this equates to an apparent fall of the body that is rapidly compensated for by an opposing motor response to maintain balance (Day et al., 1997).

The beauty of this stimulus–response is that it takes into account the spatial relationship between the head and the feet (vestibular–proprioceptive integration) (Lund & Broberg, 1983) and the availability of visual information (vestibular–visual integration) (Britton et al., 1993; Day & Bonato, 1995; Day & Guerraz, 2007; Smetanin et al., 1990). Based on the known perceptual deficits in PCA and tAD described above, we consider both integrative processes to be reasonable candidates for investigation. They were assessed by comparing the GVS-evoked responses in healthy controls with those in people with PCA and tAD under various head orientation and visual conditions. We hypothesized: (1) abnormal balance response variability and direction owing to disturbed spatial transformation of information between sensory and effector reference frames in both patient groups; and (2) abnormal modulating influence of vision on balance response magnitude owing to visual processing deficits in PCA.

**Table 1. Demographic and vibrotactile participant characteristics**

| | Control | | PCA | | tAD | |
|---|---|---|---|---|---|---|
| **(A) Demographic information** | | | | | | |
| N (male:female) | 21 | (13:8) | 18 | (11:7) | 18 | (10:8) |
| Age (years) | 69.8 | (±5.2) | 63.3 | (±6.5) | 71.4 | (±7.0) |
| MMSE (/30)[1] | na | | 22.5 | (±4.6) | 21.3 | (±5.3) |
| Amyloid PET/CSF[2] | na | | 6/6 | | 3/3 | |
| **(B) Vibrotactile threshold (µm)** | | | | | | |
| Plantar hallux | 23.5 | (13.5–37.0) | 16.0 | (13.5–21.5) | 28.0 | (17.9–37.9) |
| Plantar metatarsal head | 24.0 | (14.5–40.5) | 12.5 | (10.0–22.5) | 27.1 | (17.0–37.4) |
| Plantar heel | 27.0 | (15.0–41.0) | 15.0 | (9.5–19.5) | 28.0 | (18.5–33.4) |
| Mid-tibia | 27.0 | (25.0–34.5) | 29.0 | (14.5–40.0) | 33.5 | (21.4–38.5) |

*Note*: means (±SD) are reported for (A) demographic information and medians (1st–3rd quartile) for (B) vibrotactile characteristics of healthy controls, posterior cortical atrophy (PCA) and typical Alzheimer's disease patients (tAD).
[1]Mini Mental State Examination – maximum score shown in parentheses.
[2]Positive amyloid imaging performed as part of another investigation or CSF (A$\beta$)1–42 $\leq$ 627 and/or tau/(A$\beta$) ratio $\geq$ 0.52 consistent with AD (Alzheimer's disease).

## Methods

### Ethical approval

Written informed consent was obtained from all participants according to the *Declaration of Helsinki* except for the requirement of registration in a public database. Ethical approval for the study was provided by the National Research Ethics Service Committee London Queen Square (reference: 06/Q0512/81).

### Participants

Eighteen patients with PCA, 18 with tAD and 21 healthy controls were included in the study. PCA and tAD patients fulfilled clinical criteria for PCA-pure and research criteria for probable AD, respectively (Crutch et al., 2017; McKhann et al., 2011). All available molecular pathology was consistent with underlying AD. Groups were of comparable sex and patient groups were of comparable disease severity. The PCA group was younger than both tAD and control groups (Table 1). The effect of age on task performance was subsequently investigated; see Results for details.

### Neuro-otological, vibrotactile and neuropsychological assessments

All participants underwent vibrotactile sensitivity assessments (Table 1) as well as balance- and fall-related screening interviews. Interview questions included the presence of vertigo, dizziness, unsteadiness or oscillopsia in the last 3 months. Detailed neuro-otological assessments were carried out in a participant subset (PCA $n = 15$; tAD $n = 10$), including those who reported dizziness, vertigo or unsteadiness (PCA $n = 2$; tAD $n = 1$). These assessments included ophthalmoscopy, oculomotor assessments (fixation, saccades, spontaneous and gaze-evoked nystagmus, pursuit, vestibulo-ocular reflex testing, and positional manoeuvres), gait assessment and the Romberg test. There was no evidence of peripheral sensory loss in PCA and tAD relative to control groups based on vibrotactile sensitivity assessments (Table 1). Clinical vestibular assessments were normal in all patients assessed. However, subtle oculomotor anomalies were noted in the PCA group, such as square wave jerks, delayed saccadic reaction and reduced saccadic range. Conversely, oculomotor tests in tAD patients were generally normal, except for the presence of square wave jerks during fixation. These oculomotor irregularities align with findings from previous studies and are characteristic of PCA and Balint's syndrome, which includes oculomotor apraxia (Crutch et al., 2017; Mendez et al., 2002; Shakespeare et al., 2015).

Both patient groups underwent neuropsychological assessments evaluating memory, language, arithmetic, spelling, reading, and early visual, visuoperceptual and visuospatial processing. Consistent with their clinical presentations (PCA: visually led; tAD: memory-led), PCA patients exhibited marked corticovisual deficits, with relatively preserved episodic memory and language abilities. The tAD group showed predominant episodic memory and language impairments, with a subset also presenting corticovisual difficulties (Table 2).

### Galvanic vestibular stimulation experiments

**Experimental set-up and procedure.** Participants stood feet together on a force plate (model 9281B, Kistler, Winterthur, Switzerland), recording ground reaction

**Table 2. Background neuropsychological patient characteristics**

| Neuropsychology test | PCA | tAD |
|---|---|---|
| **Early visual function** | | |
| Visual acuity – CORVIST*: Snellen (6/9) | 6/9 | 6/9 |
| Crowding – 10 alphanumeric strings (10) | 10.0 (10.0–10.0) | 10.0 (10.0–10.0) |
| Figure-ground discrimination – VOSP[†] (20) | 17.0 (16.0–18.6) | 19.0 (19.0–20.0) |
| Shape discrimination[‡] (20) | 16.0 (10.1–18.0) | 20.0 (17.5–20.0) |
| Hue discrimination – CORVIST (4) | 3.0 (2.0–4.0) | 3.5 (3.0–4.0) |
| **Visual–perceptual function** | | |
| Fragmented letters – VOSP (20) | 12.0 (8.0–14.0) | 19.0 (17.1–19.6) |
| Object decision – VOSP (20) | 12.0 (9.5–13.6) | 17.5 (14.1–18.0) |
| Unusual views (20) | 4.5 (2.0–6.0) | 13.5 (10.1–16.0) |
| Usual views (20) | 17.5 (15.0–18.0) | 19.0 (19.0–20.0) |
| **Visual–spatial function** | | |
| Number location – VOSP (10) | 2.5 (0.0–4.6) | 9.0 (6.1–10.0) |
| Dot counting – VOSP (10) | 7.0 (6.0–8.6) | 10.0 (8.1–10.0) |
| A-cancellation – time taken to complete (90 s) | 53.5 s (42.6–62.1 s) | 28.5 s (22.5–38.5 s) |
| **Non-visual parietal function** | | |
| Calculation – GDA[§] (24) | 15.0 (10.0–17.0) | 15.0 (13.1–16.6) |
| Cognitive estimates – general knowledge (30) | 8.0 (4.5–15.6) | 10.5 (4.1–16.6) |
| Digit span – forward (12) | 6.5 (5.1–7.6) | 6.0 (6.0–7.0) |
| Digit span – backward (12) | 4.0 (3.0–5.0) | 4.0 (2.0–5.0) |
| **Memory function** | | |
| Short recognition memory test – words[¶] (25) | 22.0 (20.1–23.6) | 18.0 (15.1–20.6) |
| Short recognition memory test – faces[¶] (25) | 22.0 (19.0–23.0) | 21.5 (19.1–23.6) |
| Paired associate learning – Camden (24) | 11.0 (8.0–14.0) | 1.0 (0.0–9.0) |
| **Language function** | | |
| Concrete synonyms (25) | 21.5 (20.0–22.6) | 20.5 (19.1–23.0) |
| Spelling – GDST[#] – set B first 20 items (20) | 15.5 (10.0–18.6) | 13.0 (7.0–19.0) |
| Naming – verbal description (20) | 18.0 (15.0–19.0) | 15.5 (10.1–17.6) |

*Note*: Medians (1st–3rd quartile) are reported for raw neuropsychological scores (maximum score shown in parentheses) of posterior cortical atrophy (PCA) patients and typical Alzheimer's disease patients (tAD).
*Cortical Visual Screening Test (James & Plant, 2001).
[†]Visual Object and Space Perception Battery (Warrington & James, 1991).
[‡]Efron (Efron, 1968): oblong edge ratio 1:1.20.
[§]Graded Difficulty Arithmetic Test.
[¶]Joint auditory/visual presentation.
[#]Graded Difficulty Spelling Test.

forces at 200 Hz, while facing a fixed visual environment. Motion-capture system markers (CODA, Charnwood Dynamics, Rothley, UK) were attached over the head (×4), C7 vertebra, shoulders (left and right acromion process), pelvis (left and right superior iliac crest), sacrum and ankles to record whole-body motion at 200 Hz. Participants wore a safety harness to prevent vertical drops and digitally controlled liquid-crystal spectacles (PLATO visual occlusion spectacles, Translucent Technologies, Toronto, Ontario, Canada), which were either transparent or opaque to allow or occlude vision.

Binaural, bipolar GVS was administered at a current intensity of 1 mA via two 3 cm diameter electrodes fixed to the mastoid processes. GVS was administered over a total of 80 trials with an equal number of trials of each polarity (anode right, cathode left; anode left, cathode right) under different proprioceptive and visual conditions.

(1) Proprioceptive: 60 trials without vision were performed with the head facing approximately 45° left, straight ahead or 45° right (20 trials each position). Preceding each trial, head position was adjusted using a visual display consisting of three LED clusters spaced 45° apart on a circular arc at eye level, with each cluster comprising four LEDs arranged in a two-by-two grid on a vertical pole. Participants were directed to turn their heads to face the illuminated cluster.

(2) Vision: 20 trials with vision were performed with the head facing straight ahead.

All trials consisted of a 2 s baseline period, a 2 s period of GVS and a 2 s post-GVS period.

All conditions were randomized across trials.

## Response measures

GVS-evoked response direction and magnitude were measured from both the horizontal ground reaction force and horizontal displacement of the body at the level of the neck. Data were low pass filtered using a 4th-order Butterworth filter with cut-off frequency set at 5 Hz. Force response was measured as the change in horizontal force from 200 ms ($\pm$10 ms; mean over five data points to reduce noise) to 400 ms ($\pm$10 ms) after GVS onset, capturing the initial pulse of force that instigated the subsequent whole-body displacement. This subsequent body displacement was measured from 300 ms ($\pm$10 ms; mean over five data points to reduce noise) to 800 ms ($\pm$10ms) following GVS onset (Fig. 1*A*). Time windows for analysis were based on previous studies characterizing the temporal profile of GVS-evoked postural responses (e.g. Day & Guerraz, 2007; Fitzpatrick et al., 1994; Wardman et al., 2003). The 200–400 ms window captures the initial force response at the feet, representing early compensatory postural activity following the onset of vestibular perturbation. The 300–800 ms window reflects the later, more global body sway response, which takes longer to develop and involves whole-body postural adjustments incorporating sensory feedback.

Force and neck-level displacement responses for each trial were used to calculate mean response magnitude, direction and variability of response direction (angular deviation) for each experimental condition per participant. Each participant's mean response measures were calculated as:

$$\text{Mean response magnitude} = \frac{1}{n}\sum_{i=1}^{n}\sqrt{x_i^2 + y_i^2}$$

$$\text{Mean response direction} = \arctan\left[\left(\frac{\left(\frac{1}{n}*\sum_n(\sin\theta_i)\right)}{\left(\frac{1}{n}*\sum_n(\cos\theta_i)\right)}\right)\right]$$

$$\text{Angular deviation} = \sqrt{2(1-r)}$$

where:

$n$ = number of trials

$x_1$ = mediolateral component of the response on the $i$th trial

$y_1$ = anteroposterior component of the response on the $i$th trial

$\theta_i$ = response direction on the $i$th trial relative to the interaural line, indicating a response directed perfectly

towards the anodal ear and positive values indicating responses anticlockwise of the anodal ear (Fig. 1*C*)

$$r = \frac{1}{n}\sqrt{\left(\sum_{i=1}^{n}\cos\theta_i\right)^2 + \left(\sum_{i=1}^{n}\sin\theta_i\right)^2}$$

To determine whether GVS response direction was appropriately integrated with head position

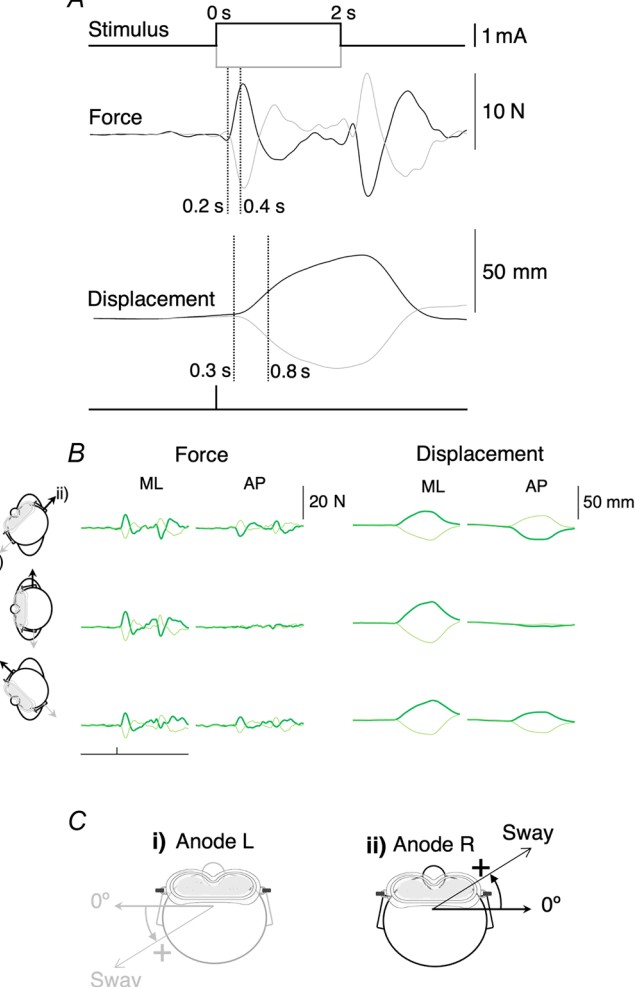

**Figure 1. GVS-evoked response magnitude and direction**
*A*, trial timing (2 s baseline, 2 s after GVS onset, 2 s after GVS offset) and mean GVS-evoked response magnitude in the control group without vision with the head straight head. GVS-evoked response magnitude is determined through mediolateral force and neck-level displacement, with measurement windows overlaid. GVS-evoked response direction convention is shown with 0° being directed towards the anodal ear (anode left: grey; anode right: bold). *B*, mean GVS-evoked medio-lateral (ML) and anterior–posterior (AP) responses in the tAD group without vision with the head right, straight ahead or left. *C*, error convention relative to the yaw head angle (interaural line), where the positive sign indicates responses anticlockwise of the anodal ear; i/ii, anode left and anode right respectively.

(proprioceptive–vestibular integration: without vision), response direction and variability (angular deviation) were measured across head positions. Responses were measured in the absence of vision for each polarity condition, averaging across all head position conditions (left, straight, right). Measures corresponded to polarity-determined (anode: left, right) GVS response direction, yielding two means of angular deviation, response direction and response magnitude per participant for both force and displacement measures.

To determine whether GVS response magnitude was appropriately modulated by visual information (visuo-vestibular integration: head straight), response magnitude was measured without and with vision. Responses were measured with the head straight, averaging across both polarity conditions. Measures corresponded to GVS response magnitude without or with vision, yielding two means of response magnitude per participant for both force and displacement measures.

### Neuroimaging: image acquisition and pre-processing

**Grey matter.** T1-weighted volumetric magnetic resonance scans were acquired on a Siemens Prisma 3T scanner using a Magnetization Prepared Rapid Acquisition Gradient Echo sequence. Images were obtained with 282 mm field of view, $256 \times 256$ acquisition matrix and the following acquisition parameters: echo time (TE) = 2.9 ms, repetition time (TR) = 2200 ms and inversion time (IT) = 900 ms.

Image pre-processing was performed using SPM12.1, involving: (1) tissue segmentation using SPM12.1's unified model (Ashburner & Friston, 2005); (2) creation of a study-specific template using geodesic shooting (Ashburner & Friston, 2011); (3) normalization of the segmentations to standard space [Montreal Neurological Institute (MNI) template] using the resulting deformations; (4) modulation to account for local volume changes; and (5) smoothing using a 6 mm full width at half maximum (FWHM) Gaussian kernel to balance the detection of small-scale anatomical differences while ameliorating misalignment.

### White matter integrity

Diffusion-weighted images (DWIs) were acquired along 64 different diffusion encoding directions using two identical single-shell GRAPPA parallel imaging spin-echo EPI sequences (SE-EPI, phase-encoding direction: anterior–posterior; resolution = 2.5 mm isotropic; TR = 6900 ms; TE = 91 ms, field of view = 240 mm; matrix = $96 \times 96 \times 55$ slices; 64 images at $b = 1000$ s/mm$^2$ and eight 'B0' images with no diffusion weighting). Gradient echo field maps (phase-encoding direction:

anterior–posterior; TE1 = 4.92 ms; TE2 = 7.38 ms; TR = 688 ms, $64 \times 64 \times 55$ slices; resolution = 3 mm isotropic; field of view = 192 mm) were acquired to correct for EPI geometric distortions arising from susceptibility.

Image pre-processing involved the following steps: (1) initial group-wise registration of B0 images to form a B0 average; (2) affine registration of the T1 image to up-sampled (factor of 2) B0 average; (3) creation of a brain mask using the Brain Extraction Tool (Smith, 2002) on the average B0 for later registration purposes; and (4) affine registration of the DWIs to the average B0 image and EPI susceptibility distortion correction using a field map-based correction technique. Voxel-wise diffusion tensor fitting of the processed and resampled DWIs was performed using the NiftyFit package (Melbourne et al., 2016). Prior to analysis images were further pre-processed using DTI-TK (Zhang et al., 2007), involving: (1) creating a high-resolution population-specific tensor template; (2) normalizing images to the template; (3) generating fractional anisotropy (FA) maps; (4) skeletonizing the template's FA (threshold: 0.2); and (5) projecting the images onto the FA skeleton. Similar steps were repeated for mean, axial and radial diffusivity maps to retain complementary information from each metric.

### Statistical analysis

**Galvanic vestibular stimulation (experiments.** All force and displacement measures were analysed using mixed-effects linear regression models with restricted maximum likelihood estimation, allowing for different variances in the three groups.

GVS response direction analyses of angular deviation and mean direction included fixed effects for group and polarity (anode left or right) and random effects for participant. GVS response magnitude analyses included fixed effects for group and vision (without or with vision) plus their interaction, and random effects for participant. For GVS response magnitude analyses (without or with vision), a log transformation was used to improve the extent to which the normality assumptions made by the model were satisfied. Results on the log-transformed scale were subsequently back-transformed to enable interpretation of results as geometric means, percentage differences and ratios of geometric means. To limit multiple testing, interaction terms were included only if there was *a priori* justification and/or evidence of improved model fit based on a likelihood ratio test.

### Neuroimaging

Voxel-based morphometry (VBM) (Ashburner & Friston, 2000) was performed to assess the relationship between smoothed, modulated and warped grey matter volume

and GVS balance responses with vision across the whole brain at a voxel-wise level, using SnPM (Nichols & Holmes, 2002). We employed SnPM's Multisubject Correlation model to examine within-group relationships between modulated grey matter volume and GVS response measures, using a multiple regression design matrix with continuous behavioural regressors. Prior to analysis a whole-brain grey matter mask was defined to include only voxels for which the intensity was greater than 0.1 in at least 90% of the images to circumvent exclusion of voxels most vulnerable to brain atrophy (Ridgway et al., 2009) and voxel variance estimates were pooled over neighbouring voxels in each scan using a 6 mm FWHM Gaussian kernel. Statistical significance was determined by permutation testing (30,000 permutations) based on peak-voxel inference and set at $P < 0.05$ (family-wise error corrected – FWE).

Whole-brain tract-based spatial statistics (TBSS) (Smith et al., 2006) was performed to investigate associations between GVS balance responses with vision and diffusion metrics in the white matter (fractional anisotropy – FA; mean diffusivity – MD; radial diffusivity – RxD; and axial diffusivity – AxD; projected on to the FA skeleton) on a voxel-by-voxel basis. Statistical significance was assessed through permutation testing (Winkler et al., 2014) (5000 permutations), and threshold-free cluster enhancement with 2D optimization was used to correct for multiple comparisons. Variance smoothing with a sigma of 2 mm (FWHM equivalent: 4.6 mm) was used to pool variance estimates over neighbouring voxels in each image.

The lower permutation number for TBSS reflects the reduced number of comparisons due to skeletonization, allowing accurate estimation of the null distribution with fewer permutations. In contrast, the higher number for VBM was chosen to ensure sufficient precision given the greater voxel-wise testing burden.

All regression models included age, sex and mini–mental state examination (MMSE) as covariates. Total intracranial volume was also included for the VBM analysis.

Correlation analyses were conducted separately within each group to avoid confounding neuroimaging analysis of GVS balance responses with clinico-radiological group differences.

We used the Harvard–Oxford cortical and subcortical structural atlases and JHU ICBM-DTI-81 white matter labels and tracts atlases, validated *in vivo* datasets and widely adopted in neuroimaging research, to label significant grey matter regions and white matter tracts, ensuring both reliability and comparability across studies.

# Results

## GVS force plate and displacement responses

**Proprioceptive–vestibular integration: GVS responses across head positions.** Mean observed GVS response direction variability (angular deviation), mean direction (angular mean) and response magnitude across head positions were comparable between control, PCA and tAD groups without vision (Fig. 2).

There was no evidence that response direction differed between groups based on measures of force (global $P = 0.29$) or displacement (global $P = 0.96$). Across all three groups, there was no evidence of an effect of polarity on force plate response direction (anode right: $-0.27°$ 95%CI $[-2.72, 2.18]$; anode left: $-0.93°$ $[-3.38, 1.52]$; $P = 0.67$). Relative to the anodal ear, neck-level displacement response was consistently directed slightly away from the inter-aural line towards the nose (anode right: $8.93°$ $[6.50, 11.35]$; anode left: $-5.53°$ $[-7.96, -3.11]$; $P < 0.001$), but there was no evidence that this effect of polarity differed between groups.

There was no evidence that angular deviation differed between groups based on measures of force (global $P = 0.88$) or displacement (global $P = 0.18$). Across all three groups, there was no evidence of an effect of polarity on angular deviation based on measures of force (estimated mean angular deviation: anode right: $33.67°$ $[30.97, 36.38]$; anode left: $34.65°$ $[31.95, 37.36]$; $P = 0.36$) or displacement (anode right: $29.07°$ $[26.35, 31.80]$; anode left: $29.10°$ $[26.37, 31.82]$; $P = 0.98$).

There was no evidence that response magnitude differed between groups based on measures of force (global $P = 0.46$) or displacement (global $P = 0.33$). Across all three groups, there was no evidence of an effect of polarity on force plate response magnitude (anode right: 6.82 N $[5.79, 7.86]$; anode left: 6.61 N $[5.61, 7.61]$; $P = 0.17$). While there was evidence that neck-level displacement response magnitude was slightly greater when the anode was on the right (anode right: 15.75 mm $[13.87, 17.63]$; anode left: 15.03 mm $[13.24, 16.83]$, $P = 0.018$), there was no evidence that this effect of polarity differed between groups.

## Visuo-vestibular integration: effect of vision on GVS response

In the absence of vision, GVS responses were consistently directed towards the anodal ear (0° – Figs 1 and 3) based on force plate responses or neck-level displacement across all groups. There was no evidence of group differences in responses across head positions, suggesting comparable processing of vestibular information and

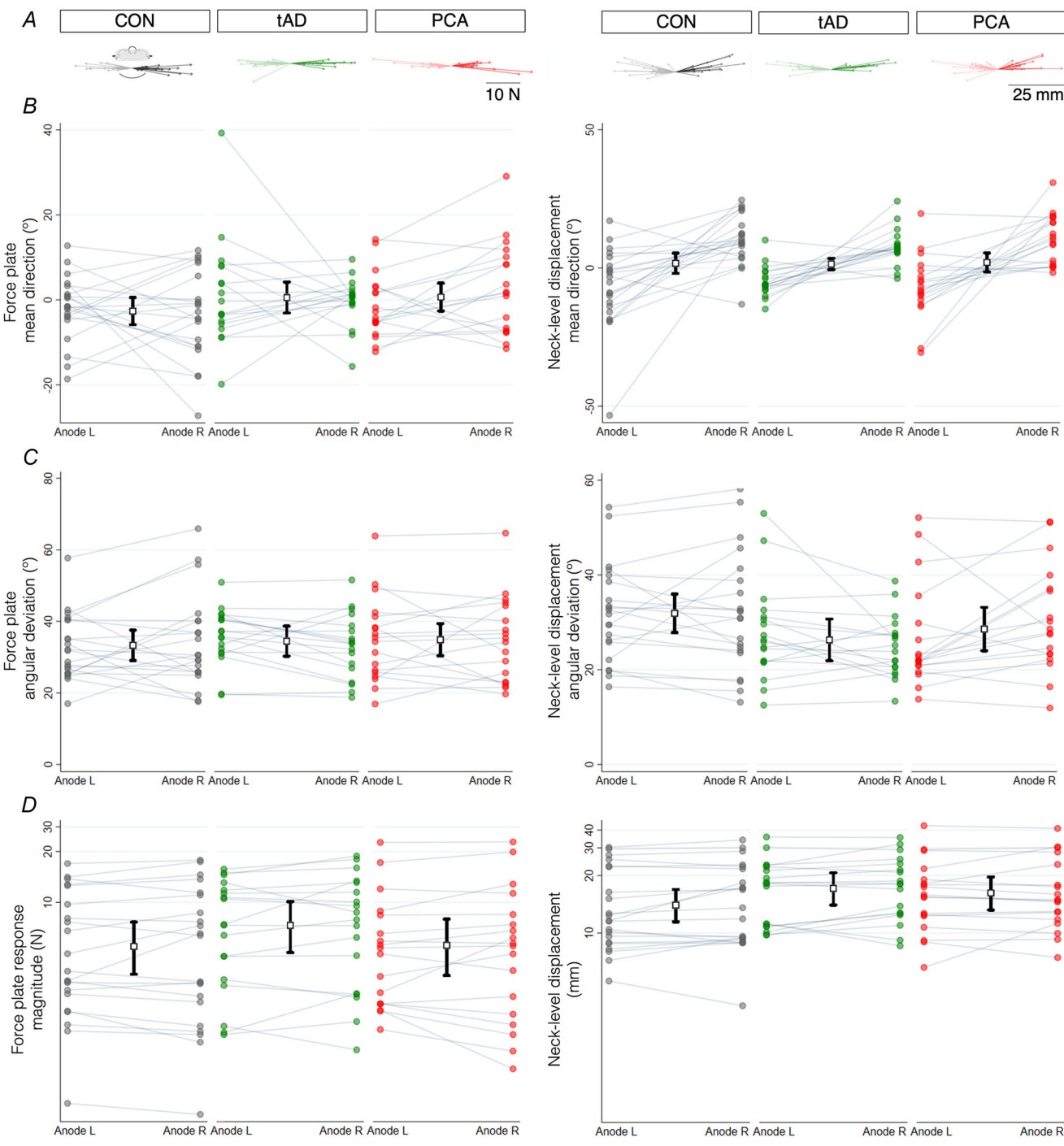

**Figure 2. Mean observed GVS responses without vision across head positions in control, tAD and PCA groups**

Mean observed GVS response (*A*), magnitude and direction (anode left: grey; anode right: bold), (*B*) direction, (*C*) variability (angular deviation) and (*D*) magnitude. Responses are averaged across head positions (left, straight, right) without vision. Responses are determined by force plate (left) and neck displacement (right). *A*, response direction averaged across head positions with responses being directed towards the anodal ear; *B* and *C*, error convention whereby 0° corresponds to a response directed towards the anodal ear and positive values indicate responses anticlockwise of the anodal ear (Fig. 1*C*).

proprioceptive–vestibular integration between all three groups.

Observed force responses were reduced in all groups with compared to without vision, consistent with the ability to modulate balance response with available visual information. However, the effect of vision on observed force response was smaller in PCA relative to control and tAD groups (Fig. 3 and Table 3).

Across all three groups, there was evidence that force response magnitude was greater without compared to with vision (estimated geometric mean magnitude [95% CI]: without vision: 6.52 N [5.61, 7.42]; with vision: 2.56 N [2.20, 2.91]). While this increase was twofold in control and tAD groups (ratio = 2.05 and 2.15, Table 3*A*), the modulating effect of vision was reduced in PCA (ratio = 1.58) with responses with vision being 47% and 57% greater than control and tAD groups respectively (Table 3*B*). Formal tests of differences between inter-

action terms provided evidence that the effect of vision in modulating force plate responses was smaller in PCA relative to both control and tAD groups (PCA *vs* controls: $P = 0.002$; PCA *vs* tAD: $P < 0.001$; global $P < 0.001$). There was no evidence that the effect of vision on force plate response differed between tAD and control groups ($P = 0.48$).

Across all three groups, there was evidence that neck-level displacement magnitude was greater without compared to with vision (estimated geometric mean magnitude [95% CI]: without vision: 14.56 mm [12.97, 16.15]; with vision: 6.17 mm [5.49, 6.83]). This increase was roughly twofold in control and tAD groups but reduced in PCA (Table 3*A*) with responses with vision being 31% and 53% greater than control and tAD groups respectively (Table 3*B*). Formal tests of differences between interaction terms provided evidence that the effect of vision in modulating neck-level displacement

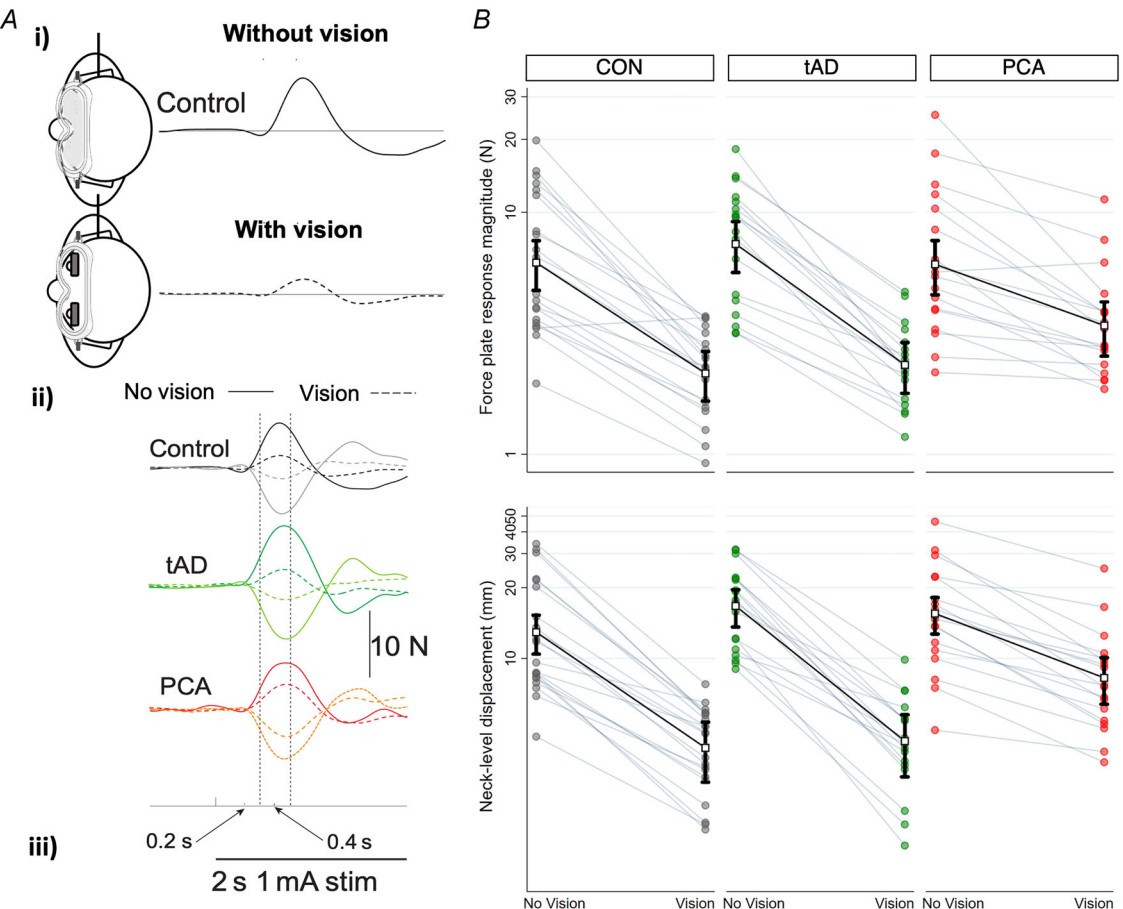

**Figure 3. Effect of vision on GVS response magnitude in control, tAD and PCA groups; response magnitude is shown without and with vision with the head straight ahead**
*A*, mean observed horizontal medio-lateral component of force acting on the body without and with vision (i: mean measured control responses with anode right; ii: full time-course of mean responses for all groups; iii: earliest component of response and measurement window). Full and dashed lines show responses without and with vision, and darker and lighter colours show responses with anode right and left respectively. *B*, response magnitude determined by force plate (top) and neck-level displacement (bottom), averaged across polarity conditions. Mean observed response magnitude is presented with estimated marginal means and 95% CIs overlaid in black.

**Table 3. Estimated geometric mean response magnitude comparing vision conditions within and between groups expressed as ratios**

**(A) Main effect of vision**

| | Control | | PCA | | tAD | |
|---|---|---|---|---|---|---|
| **Response magnitude** | | | | | | |
| Force plate response | 2.05 | (1.86, 2.25) | 1.58 | (1.37, 1.80) | 2.15 | (1.97, 2.33) |
| Neck-level displacement | 1.89 | (1.73, 2.04) | 1.58 | (1.45, 1.71) | 2.11 | (1.94, 2.27) |

**(B) Comparing vision effect between groups**

| | PCA *versus* Control | | PCA *versus* tAD | | tAD *versus* Control | |
|---|---|---|---|---|---|---|
| **Response magnitude** | | | | | | |
| Force plate response | 0.53 | (0.24, 0.82) | 0.43 | (0.15, 0.72) | 1.10 | (0.83, 1.36) |
| Neck-level displacement | 0.69 | (0.49, 0.90) | 0.47 | (0.27, 0.68) | 1.22 | (1.00, 1.44) |

*Note*: Estimated geometric mean force and displacement responses expressed as ratios comparing (A) visual conditions (without *vs* with vision) within each group, and (B) effects of vision between groups. 95% CIs are in parentheses.

**Table 4. Grey matter correlates of PCA GVS response magnitude with vision – peak voxels**

| $k$ | $P_{FWE}$ | $x$ | $y$ | $z$ | Mean | ($\pm$SEM) | Peak grey matter region |
|---|---|---|---|---|---|---|---|
| **(A) GVS-evoked force plate response with vision** | | | | | | | |
| 23 | 0.032 | 30 | −88 | −14 | 0.49 | ($\pm$0.05) | right Occipital fusiform gyrus |
| | 0.044 | 22 | −92 | −14 | | | right Occipital pole |
| 13 | 0.023 | 08 | −20 | 3 | 0.53 | ($\pm$0.03) | right Thalamus |
| 12 | 0.039 | 09 | −90 | −10 | 0.56 | ($\pm$0.07) | right Lingual gyrus |
| **(B) GVS-evoked neck displacement response with vision** | | | | | | | |
| 32 | 0.009 | 8 | −20 | 2 | 0.51 | ($\pm$0.03) | right Thalamus |
| 2 | 0.043 | 14 | −64 | 12 | 0.56 | ($\pm$0.04) | left Intracalcarine cortex |

*Note*: Peak voxel information for significant clusters showing lower grey matter volume associated with (A) greater GVS-evoked force plate responses with vision and (B) greater GVS-evoked neck displacement responses with vision in PCA. For each cluster, peak MNI coordinates (*x,y,z*), cluster size (*k*), family-wise error corrected *P*-value at $P < 0.05$ ($P_{FWE}$) and mean ($\pm$SEM) of grey matter volume are reported. Abbreviations: GVS, galvanic vestibular stimulation.

responses was smaller in PCA relative to both control and tAD groups (PCA *vs* controls: $P = 0.003$; PCA *vs* tAD: $P < 0.001$; global $P < 0.001$). There was a tendency towards a greater effect of vision in tAD relative to the control group, but this was not statistically significant ($P = 0.052$).

*Post hoc* analyses provided evidence of a statistically significant association between age and neck-level displacement, but not force response across all groups. Age-adjusted neck-level displacement response to GVS was greater without compared to with vision (estimated geometric mean magnitude [95% CI]: without vision: 14.56 mm [13.10, 16.02]; with vision: 6.17 mm [5.50, 6.78]). Formal tests of differences between interaction terms provided evidence that the effect of vision in reducing neck-level displacement responses was smaller in PCA relative to both control and tAD groups (PCA *vs* controls: $P = 0.003$; PCA *vs* tAD: $P < 0.001$; global

$P < 0.001$). When adjusting for age, there was a nominally statistically significant greater effect of vision in tAD relative to the control group (tAD *vs* controls: $P = 0.043$).

### Neuroimaging

**VBM analysis – grey matter.** In the PCA group, greater GVS-evoked force plate responses with vision were associated with lower grey matter (GM) volume in several right-hemispheric dominant clusters (Table 4*A*). The largest cluster was located in the occipital fusiform gyrus ($P = 0.032$) and extended posteriorly into the occipital pole ($P = 0.044$). Smaller clusters were also found in the thalamus and lingual gyrus ($P = 0.039$, FWE-corrected) (Fig. 4*A*). Greater GVS neck-level displacement responses with vision were associated with lower GM volume in two clusters (Table 4*B*), involving the right thalamus ($P = 0.023$) and the left intracalcarine cortex ($P = 0.043$)

(Fig. 4*B*). No statistically significant associations were found between GVS responses and GM volume in the tAD group after correcting for multiple testing.

**TBSS analysis – white matter.** In the PCA group, greater GVS force plate responses with vision were associated with increased FA in two clusters. The largest cluster was in the right hemisphere ($P = 0.029$) and included the retrolenticular part of the internal capsule, anterior and posterior corona radiata, posterior thalamic radiation, anterior and posterior limb of the internal capsule, superior corona radiata, external capsule, and the splenium of the corpus callosum. The second cluster was in the left hemisphere ($P = 0.040$) and included the anterior and posterior limb of the internal capsule, superior corona radiata and external capsule (Fig. 5*A*). Associations between higher FA and greater neck-level displacement responses with vision were observed in one cluster ($P = 0.042$) located in the right hemisphere and included the posterior and superior corona radiata (Fig. 5*B*). In the tAD group, there were no statistically significant associations between balance responses and white matter integrity after correcting for multiple testing.

## Discussion

We assessed vestibular-evoked balance responses under varying head position and visual conditions in PCA, tAD and control participants. Averaging across head positions without vision, GVS-evoked responses were comparable between the groups. Averaging across polarity conditions with the head facing forward, GVS-evoked response magnitude was smaller in all three participant groups with vision than without. However, this attenuating effect of vision was smaller in PCA relative to tAD and control groups, and was associated with occipito-thalamic atrophy and diffuse white matter abnormalities. These findings support our hypothesis of disturbed visual influence on the vestibular control of balance in PCA, but do not lend support to a disturbed spatial transformation of information between sensory and effector reference frames. First, we discuss the visuo-vestibular disturbance in PCA and its potential underlying mechanisms.

### Visuo-vestibular integration for balance

Previous studies in healthy individuals have demonstrated the phenomenon of visual modulation of vestibular-evoked balance responses. Visual information, varying from a three-dimensional scene down to a single-point light source, acts to attenuate the GVS-evoked balance response (Day & Bonato, 1995; Day & Guerraz, 2007; Smetanin et al., 1990). This effect of vision has been attributed to two separate mechanisms involving feedback and feedforward processes (Day & Guerraz, 2007). The feedback mechanism arises from the optic flow caused by the GVS-evoked balance response itself, which causes the body to sway and the head to move

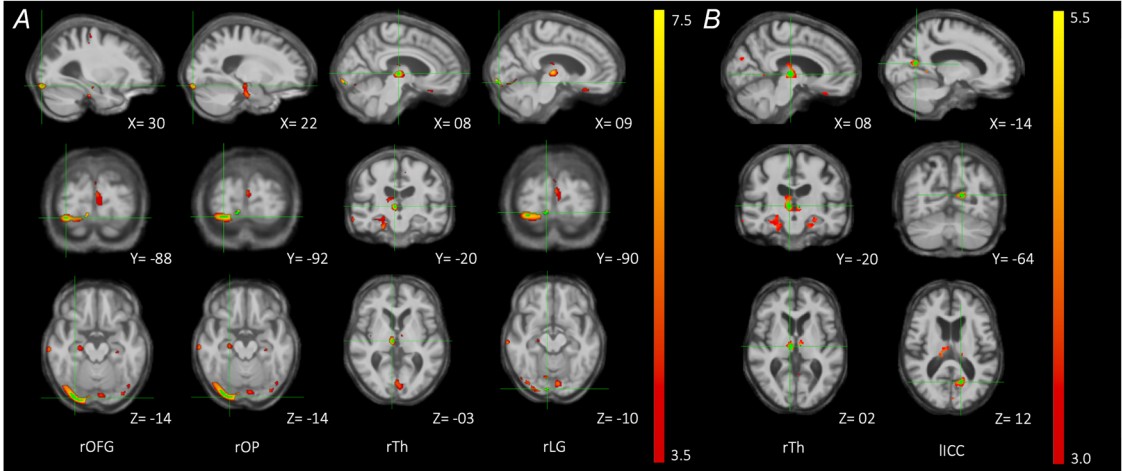

**Figure 4. Grey matter correlates of PCA GVS response magnitude with vision determined by whole-brain voxel-based morphometry**
Statistical parametric maps (SPMs) of associations between lower regional grey matter volume and greater response magnitude with vision determined by (*A*) force response and (*B*) neck displacement in the posterior cortical atrophy group. Family-wise error-corrected significant (*P* < 0.05) regions, identified by permutation-based peak-voxel inference (30,000 permutations), are shown in green and annotated by cross-hairs. SPMs are thresholded at *P* < 0.001 (uncorrected) for visualization purposes and overlaid on a normalized study-specific T1-weighted group template. Colour bar represents pseudo t-scores. X,Y,Z coordinates indicate the slice position in millimetres in MNI space. OFG, right fusiform gyrus; rOP, right occipital pole; rTH, right thalamus; rLG, right lingual gyrus; lICC, left intracalcarine cortex.

in space. It was shown that such feedback effects have little impact before 400 ms after GVS onset; earlier effects are therefore probably due to a feedforward process in which the gain of the vestibulo-spinal loop is modulated by the current visual input (Day & Guerraz, 2007). This modulating effect of vision was reduced in PCA relative to control and tAD groups so that force-plate responses were 47–57% greater with vision. The smaller attenuating effect of vision observed here in the PCA group was evident during the earliest mechanical indicator of GVS responses detected by the force plate between 200 and 400 ms following stimulus onset (Fig. 3). Thus, this decreased effect of vision cannot exclusively be attributed to feed-back effects but suggests feedforward reductions in the weighting of visual input attenuating vestibular-evoked balance response. Feedforward re-weighting mechanisms may rapidly and dynamically adjust to available visual information in healthy participants or slowly owing

to chronic sensory loss (Day & Cole, 2002; Day & Guerraz, 2007). In the current PCA group, feedforward reduction of visual input from cortex onto presumed sub-cortical vestibulo-spinal pathways may itself arise from progressive deterioration of visual motion processing, discussed below with reference to neuroanatomical substrates.

Neuroanatomical findings indicate possible cortico-subcortical pathways subserving visual contributions to vestibular balance control. Within the PCA group, the reduced modulating effect of vision on GVS-evoked responses was correlated with volume loss within ventral occipito-temporal cortices, reduced thalamic volume, and abnormalities of contiguous retrolenticular and posterior radiations. These cortico-thalamic regions have functional characteristics and reciprocal connections relevant for processing and integrating visual and vestibular information.

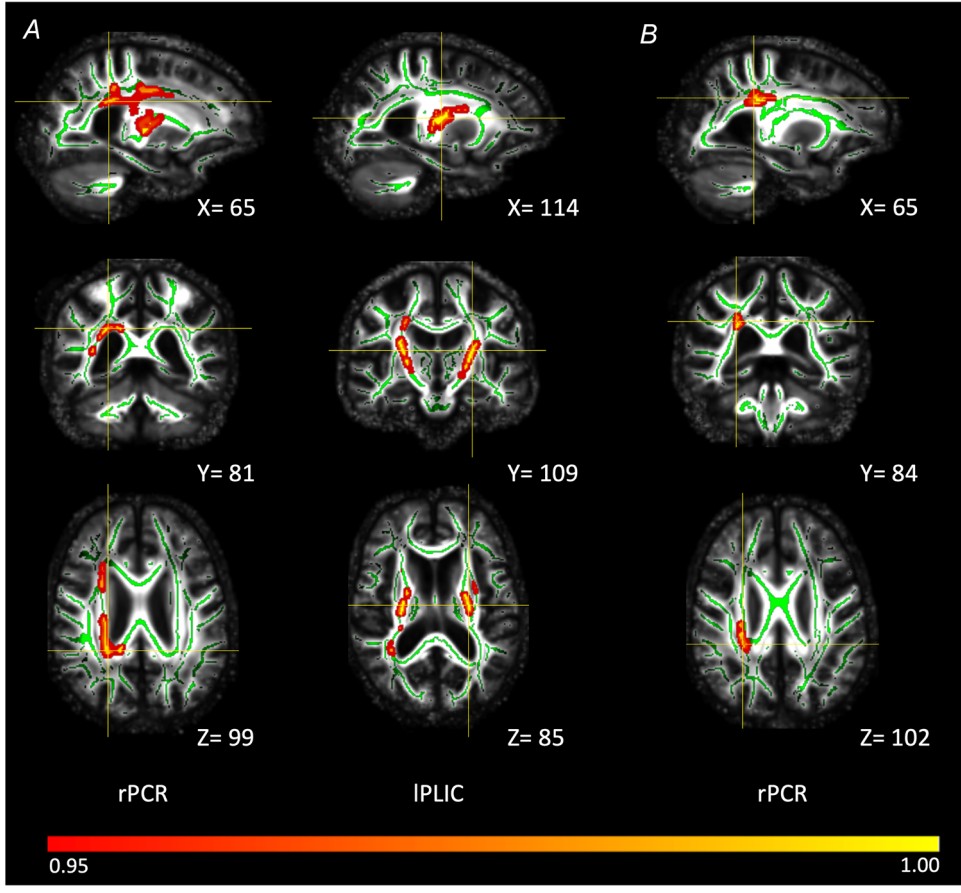

**Figure 5. White matter correlates of PCA GVS response magnitude with vision determined by whole-brain tract-based spatial statistics**

Associations between higher fractional anisotropy and greater response magnitude with vision determined by (*A*) force response and (*B*) neck displacement in the posterior cortical atrophy group. Displayed are significant peak clusters ($P < 0.05$ – FWE-corrected), with crosshairs marking peak voxels at corresponding MNI (X,Y,Z) coordinates. Significant voxels are shown in red–yellow, and white matter skeleton voxels are displayed in green and overlaid onto mean fractional anisotropy computed from all subjects. Colour bar represents the range of *P*-values in the overlaid clusters. rPCR, right posterior corona radiata; lPLIC, left posterior limb of the internal capsule.

Occipito-temporal correlates overlap with neuro-anatomical correlates of self-motion detection, sensori-motor transformations necessary for movements such as grasping, visual motion discrimination and visual crowding in PCA, relevant for the fidelity of visual contributions to balance (Fischer et al., 2012; Orban et al., 1998; Ruehl et al., 2022; Schlack et al., 2005; Yong et al., 2014). Thalamic–cortical correlates may be involved both in faulty visual processing and multisensory convergence. For example, the paracentral nuclei receive inputs from occipitotemporal regions (Van Der Werf et al., 2002) outlined above, while pulvinar activation may be induced by both optokinetic and vestibular stimulation (Bottini, 2001; Dieterich & Brandt, 2018; Lopez & Blanke, 2011; Saalmann et al., 2012). Previous human brain activation studies have demonstrated intimate and inhibitory interactions between visual and vestibular cortical regions, including visual cortical responses to optokinetic stimulation accompanied by deactivation of vestibular cortices. These findings suggest disruption to a cortico-thalamic network inhibiting vestibular-driven balance response based on conflicting visual information.

### Proprioceptive–vestibular integration for balance

Accurate interpretation of vestibular cues and integration with other sensory information, such as that signalling head orientation relative to the feet, are crucial for appropriate GVS-evoked balance responses (Fitzpatrick & Day, 2004). The comparable GVS responses obtained without vision across different head positions and groups indicate reliable processing of vestibular information as well as normal proprioceptive–vestibular integration in both patient groups. Thus, we were unable to reject the null hypothesis that response magnitude, direction and directional variability do not differ between patient and control groups.

This is an interesting null result given the clear disturbance in *perception* of verticality observed in the same participants (Day et al., 2022). The *perceptual* disturbance was present irrespective of whether verticality was judged visually or haptically. It was argued that the disturbance arose from the mechanism that spatially transforms graviceptive information between different reference frames, possibly due to posterior parietal vulnerability. Likewise in the present experiments, the graviceptive information from the vestibular system had to be spatially transformed from its craniocentric reference frame to one in which the leg muscles operate. Thus, the leg muscles reacted differently to the fixed vestibular stimulus to produce variable body motion that was locked to the head's orientation. The implication is that the spatial transformation of graviceptive information for perception of verticality uses different neural machinery to that for balancing the upright body. This would be compatible with proprioceptive–vestibular integration mechanisms operating subcortically for the control of balance.

### Conclusions

The apparent difference in the way PCA-related neuro-degeneration affected balance and perception implies separate systems for the two functions: one responsible for rapid self-motion detection and motor control, the other for perceiving the world in a stable, upright manner (Day et al., 2022). Presumably, the balance control system includes some of the subcortical regions relatively spared in PCA and tAD, such as second-order vestibular neurons of the vestibular nuclei, vestibular and somatosensory regions of the thalamus, the pontomedullary reticular formation and cerebellum. However, the disturbed visual modulation of balance responses in PCA suggest that some cortical regions with visual processing capabilities do interact with the balance control system. Our imaging data implicate occipitotemporal cortices, thalamic regions such as the paracentral nuclei and pulvinar, and their reciprocal connections.

There is one other curious difference between the current and previous findings. As we have shown here, the impact of vision on vestibular balance mechanisms is reduced. However, the influence of visual cues (tilted frame) on perception of verticality was shown to have an *exaggerated* effect in the same group of PCA participants (Day et al., 2022). This may be related to the different functions under investigation (perception *vs* balance) or possibly the different types of visual information relevant to each function (static *vs* dynamic visual cues). Discrepancies between perceptual and balance functions align with prior work demonstrating divergence between perceived orientation and vestibular-evoked balance responses (Dalton et al., 2017; Wardman et al., 2003). Further work is required to understand better the mechanisms underlying these phenomena.

Our findings and interpretations require several caveats. First, patient participant numbers in GVS and imaging analyses were relatively small and may limit the generalizability of our findings. Second, our patient groups were of comparable disease severity but not age (PCA: younger). However, elevated GVS responses with vision in PCA are unlikely to be attributed to younger age. While younger age may account for lower vibrotactile thresholds on plantar sites in the PCA group, further work might evaluate somatosensory balance control (e.g. effect of light touch with/without vision) to investigate the possibility of altered visual–somatosensory integration. Third, molecular/pathological data were available from a patient subset. Fourth is the absence of

neuropsychological measures specifically assessing visual motion sensitivity or specificity. Future studies aiming to characterize the role of visual motion processing in PCA or related conditions would benefit from including targeted tests of optic flow perception, motion coherence or direction discrimination. Lastly, DTI is limited in its ability to model complex multi- and crossing-fibre tracts resulting in paradoxical increases in FA values as observed in this study. These may reflect loss of crossing fibres rather than increased white matter integrity, given reports of similar increases in AD and Huntington's disease (Douaud et al., 2009, 2011; Teipel et al., 2014).

Future investigations may investigate balance training predicated on relatively reliable vestibulo-motor systems. The findings extend investigations of vertical plane disorientation (Day et al., 2022) and demonstrate the resilience and adaptability of the balance system despite neurodegenerative disease.

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

## Additional information

### Data availability statement

All data supporting the results presented here are available from the corresponding author upon reasonable request.

### Competing interests

The authors declare no competing interests.

### Author contributions

B.D., S.J.C., N.S.R., D.K. and K.X.X.Y. contributed to the conception and design of the study. D.O., B.D., M.J.B., A.P., D.C., D.K. and K.X.X.Y. contributed to the acquisition and analysis of the data. D.O., B.D., D.C. and K.X.X.Y. contributed to drafting the figures. All authors contributed to drafting the paper.

### Funding

This project was funded by an Alzheimer's Society project grant (AS-PG-14-022). D.O. has received funding from the European Research Council under the European Union's Seventh Framework Programme (FP7/2007-2013)/ERC grant agreement no. 616 905. The work was also supported by an Alzheimer's Research UK Senior Research Fellowship and ESRC/NIHR (ES/L001810/1) grant to S.C. NSR is supported by a University of London Chadburn Academic Clinical Lectureship. K.Y. is an Etherington PCA Senior Research Fellow and is funded by the Alzheimer's Society, grant number 453 (AS-JF-18-003) and the Vivensa Foundation [grant number ARHVF2402\2]. The Dementia Research Centre is supported by Alzheimer's Research UK, Brain Research Trust and The Wolfson Foundation. This work was also supported by the NIHR Queen Square Dementia Biomedical Research Unit, and the NIHR UCL/H Biomedical Research Centre and the UK Dementia Research Institute at UCL, which receives its funding from UK DRI Ltd, funded by the UK Medical Research Council, Alzheimer's Society and ARUK.

### Acknowledgements

We are deeply indebted to all the research participants and their supporters without whom this work would not have been possible. We are also grateful to members of the PCA Support Group (www.raredementiasupport.org/posterior-cortical-atrophy/) who inspired the study, as described in Crutch et al. (2018). We are also grateful for clinical and technical advice from Professor Jonathan Schott, Dr Aida Suarez Gonzalez, Dr Christopher Dakin and Dr Rebecca St George.

### Keywords

Alzheimer's disease, galvanic vestibular stimulation, multisensory processing, posterior cortical atrophy, postural control, standing balance

### Supporting information

Additional supporting information can be found online in the Supporting Information section at the end of the HTML view of the article. Supporting information files available:

**Peer Review History**

