## [Peer Review History · The Journal of Physiology]

Visual modulation of vestibular-evoked balance response disturbed by posterior cortical atrophy

Dilek Ocal, Brian L Day, Amy Lynne Peters, Matthew Bancroft, David M Cash, Diego Kaski, Natalie Ryan, Sebastian Crutch, and Keir Yong

DOI: 10.1113/JP288693

Corresponding author(s): Keir Yong (keir.yong@ucl.ac.uk)

The following individual(s) involved in review of this submission have agreed to reveal their identity: Peter zu Eulenburg (Referee #1)

Review Timeline:

Submission Date:	05-Feb-2025
Editorial Decision:	08-Apr-2025
Revision Received:	07-Aug-2025
Editorial Decision:	08-Sep-2025
Revision Received:	16-Sep-2025
Accepted:	18-Sep-2025

Senior Editor: Vaughan Macefield

Reviewing Editor: Luke Henderson

Transaction Report:

Dear Dr Yong,

Re: JP-RP-2025-288693 "Visual modulation of vestibular-evoked balance response disturbed by posterior cortical atrophy" by Dilek Ocal, Brian L Day, Amy Peters, Matthew Bancroft, David Cash, Diego Kaski, Natalie Ryan, Sebastian Crutch, and Keir Yong

Thank you for submitting your manuscript to The Journal of Physiology. It has been assessed by a Reviewing Editor and by 2 expert referees and we are pleased to tell you that it is potentially acceptable for publication following satisfactory major revision.

REVISION CHECKLIST:

We look forward to receiving your revised submission.

Yours sincerely,

Vaughan Macefield
Senior Editor
The Journal of Physiology

REQUIRED ITEMS

- Author photo and profile. First or joint first authors are asked to provide a short biography (no more than 100 words for one author or 150 words in total for joint first authors) and a portrait photograph. These should be uploaded and clearly labelled together in a Word document with the revised version of the manuscript. See Information for Authors for further details.

- The Journal of Physiology funds authors of provisionally accepted papers to use the premium BioRender site to create high resolution schematic figures. Follow this link and enter your details and the manuscript number to create and download figures. Upload these as the figure files for your revised submission. If you choose not to take up this offer, we require figures to be of similar quality and resolution. If you are opting out of this service to authors, state this in the Comments section on the Detailed Information page of the submission form. The link provided should only be used for the purposes of this submission. Authors will be charged for figures created on this premium BioRender account if they are not related to this manuscript submission.

- Please upload separate high-quality figure files via the submission form.

- Please ensure that the Article File you upload is a Word file.

- Papers must comply with the Statistics Policy: https://jp.msubmit.net/cgi-bin/main.plex?form_type=display_requirements#statistics.

In summary:

- If $n \leq 30$, all data points must be plotted in the figure in a way that reveals their range and distribution. A bar graph with data points overlaid, a box and whisker plot or a violin plot (preferably with data points included) are acceptable formats.

- If $n > 30$, then the entire raw dataset must be made available either as supporting information, or hosted on a not-for-profit repository, e.g. FigShare, with access details provided in the manuscript.

- 'n' clearly defined (e.g. x cells from y slices in z animals) in the Methods. Authors should be mindful of pseudoreplication.

- All relevant 'n' values must be clearly stated in the main text, figures and tables.

- The most appropriate summary statistic (e.g. mean or median and standard deviation) must be used. Standard Error of the Mean (SEM) alone is not permitted.

- Exact p values must be stated. Authors must not use 'greater than' or 'less than'. Exact p values must be stated to three significant figures even when 'no statistical significance' is claimed.

- Please include an Abstract Figure file, as well as the Figure Legend text within the main article file. The Abstract Figure is a piece of artwork designed to give readers an immediate understanding of the research and should summarise the main conclusions. If possible, the image should be easily 'readable' from left to right or top to bottom. It should show the physiological relevance of the manuscript so readers can assess the importance and content of its findings. Abstract Figures should not merely recapitulate other figures in the manuscript. Please try to keep the diagram as simple as possible and without superfluous information that may distract from the main conclusion(s). Abstract Figures must be provided by authors no later than the revised manuscript stage and should be uploaded as a separate file during online submission labelled as File Type 'Abstract Figure'. Please also ensure that you include the figure legend in the main article file. All Abstract Figures should be created using BioRender. Authors should use The Journal's premium BioRender account to export high-resolution images. Details on how to use and access the premium account are included as part of this email.

EDITOR COMMENTS

Reviewing Editor:

In addition to the reviewers' comments, the authors need to provide a table listing the VBM analysis results - MNI coordinates of each cluster, z value, cluster size as well as the mean SEM grey matter density values at a minimum. The authors also need to describe the second level statistical approach in more detail. The figures need to be improved by adding slice locations and cluster location labels.

Senior Editor:

Thank you for submitting your manuscript to The Journal of Physiology. I have now received comments from two independent reviewers and the Reviewing Editor, all experts in the field. As you will see from their comments, there are major issues you will need to address before we can consider the manuscript further. I invite you to revise the manuscript accordingly and submit point-by-point responses to the reviewers' comments. I look forward to receiving your revised manuscript in due course.

REFEREE COMMENTS

Referee #1:

Summary:

The authors report on group differences between healthy controls, Alzheimer and PCA patients around a GVS-induced postural response. Concurrent neuroimaging data was used to interpret the findings. The study is robust and sound, its findings though unsurprising. My addressable concerns are with respect to some methodological details, the framing of the findings and the discussion. In chronological order:

Title:

- In my interpretation of the results from the perspective of a PCA patient to the artificial vestibular stimulus is not an aggravating disturbance of the postural response but rather an attenuation due to a reduced visual motion processing in these patients. I would suggest to rephrase it a bit.

Key points:

1) All sensory information processing contains noise. Why stress it here for the multisensory cues when there is no evidence for extraordinary noise in the involved systems compared with others. Please drop the noise implication.

2) PCA as a disease is not known for gait disturbances or an increased fall risk in the clinical neurology literature. PCA is not known for lateropulsion, Pusher syndrome or loss of trunc uprightness at all. Please give references for this in the introduction or drop the point. Lesions studies only implicate the lateral thalamus and the superior temporal gyrus in verticality perception and not the parietal lobe (see works by Karnath, Baier and Perennou).

3) Short-duration GVS in primates induces a stimulation of semicircular canals and otolith organs alike inducing a head tilt in

the frontal plane (PMID: 31015434). Not a head rotation. Please correct.

4) The grey matter loss in PCA patients affect the entire dorsal visual stream. The vestibular integration of visual motion information via area 3A, 6 into area CSv for egomotion detection (see Figure 5a of the manuscript (PMID: 21709176 PMID: 36334557) and the intraparietal sulcus (VIP) for grasping (PMID: 15872109). This topic should be addressed also with respect to the previous work in this journal. The interesting findings in PCA also shine a methodological light on how we measure verticality perception visually and haptically and the role of the parietal cortex.

Methods:

1) Table 1 shows a significant difference in the vibrotactile results (the unit is unclear from the caption though) from all plantar testing points reflecting a higher sensitivity to proprioceptive input from the foot. This finding is not sufficiently addressed and discussed in the manuscript. Better proprioception from the feet to counterbalance loss of sensitivity in visual motion processing?

2) None of the neuropsychological tests include testing for visual motion sensitivity and specificity. Please address.

3) GVS stimulus: Length of rectangular stimulus unclear. Why does the electrical current onset and its offset not represent two separate events that should be modelled (PMID: 36732070)?

4) Neuroimaging analysis: Great application of permutation testing via SnPM to produce robust findings in small cohorts.

Results:

1) 3.2.1: Why not include the white matter findings from the VBM analysis to complement and support the TBSS analysis?

2) Throughout the neuroimaging findings the anatomical localization reporting for the grey matter and the fiber tracts appear a little outdated (see <https://www.ebrains.eu/tools/human-brain-atlas>) and might be improved.

Referee #2:

Ocal et al. investigated the vestibular control of balance in persons with posterior cortical atrophy and typical Alzheimer's disease. Using galvanic vestibular stimulation, they showed that the precision and accuracy of vestibular-evoked responses did not differ between group but that the responses evoked with eyes open were larger in participants with posterior cortical atrophy. The authors interpreted these findings as evidence of deficits in visual processing and visual-vestibular pathways in persons with posterior cortical atrophy. Overall, the paper is well written but there are a number of points the authors should consider.

Main comments

1- The authors mentioned in the first paragraph of the Introduction the presence of a 'coherent perception of body orientation and movement in space' to justify their study. However, it is not clear why the authors developed these arguments as there are numerous examples in the literature where perception and action don't line up - some even specific to the vestibular control of balance (see Dalton et al. 2017 & Wardman et al. 2003). Consequently, the authors should present a balanced perspective in the Introduction to justify their experiment and hypotheses. Also, their results should be interpreted in line with the known distinctions between balance actions and perception in the Discussion.

2- Interpretation of visual modulation. The authors observed larger GVS-evoked responses in participants with posterior cortical atrophy when vision was present. These results may indicate a higher gain of the vestibular channel in this group of participants. The authors, however, interpret these results as an indication of a reduction in the weighting of visual inputs onto vestibulo-spinal pathways. To accept this argument, one would have to present evidence that the gain of other sensory channels did not change in these participants, or support this statement with a computational model of the system.

Other comments

1- Related to the first major comment, the authors should consider citing the proper literature throughout the text. For example, Fitzpatrick & Day (2004) did not record vestibular afferents (page 5) but proposed a model to understand the effects of GVS. Please cite the original physiological papers (see Goldberg et al. 1984 & Kwan et al. 2019). Similarly, cite the original papers related to the craniocentricity of vestibular responses (Lund & Broberg, 1983).

2- The methods section requires additional information:

A) Where were the CODA markers located on the pelvis?

B) Were conditions randomized between participants or were the conditions performed in the order presented in the paper (page 9)? Please justify if there was no randomization.

C) The GVS amplitude was 1 mA and participants were exposed to 10 trials per condition/polarity combination. Please show the variability in the responses within and between participants given the low number of trials and small GVS magnitude may lead to variation in responses, particularly in the groups tested. The authors need to convince the reader that the evoked force and movement responses had high signal to noise ratios.

D) P10. The authors should justify why they used constant time points to determine the GVS-evoked responses.

E) Show your equations for determining the direction and angular deviation of the GVS-evoked responses. The authors mentioned a 0 deg reference system for anodal but this is not clear as it would lead to a head fixed coordinate system. Please show your reference system in Fig 1 and include both the AP and ML responses so the reader can visualize how the angular values are computed. Also represent the vector of the evoked responses in Fig 2 to show both the response magnitude and direction for all participants and conditions.

F) Why did the authors average across polarities for the vision comparison but not for the craniocentric experiment?

G) Statistics. Please justify why the authors did not use circular statistics. How did they deal with the 0-360 deg (or -180-180 deg) discontinuities in the angular data?

H) Statistics. Explicitly state what drove the non-normality in the data. This is particularly important given the main result is related to the vision-no-vision comparison.

I) Statistics. The authors mentioned interactions were included if there 'was evidence of improved model fit'. Please justify why the interaction terms were not always included in the statistical model.

J) Why did the authors change the number of permutations (5000 vs 30000) between the neuroimaging tests?

3- Angular direction of the GVS evoked responses. The authors reported values between -0.27 and -0.93 deg for the anode right and anode left conditions (in text) and Figure 2 shows a near zero angle across conditions. Shouldn't the values between anode right and left show a near 180 deg difference? This point goes back to the Methods points above.

4- Results, page 17. Please describe how the interaction terms were decomposed and clearly show the results for the interaction.

5- Results, correlation and neuroimaging. The authors should clearly justify why they performed the correlations with both the force and position signals. What are the specific hypotheses or physiological mechanisms tested? Also, clearly justify why were the correlation analyses only performed within the groups of participants?

Minor comments:

1- Page 7. One sentence (starting with 'Conversely,...') is mostly repeated twice.

2- Page 7. Consider presenting the main highlights or the tests performed and the associated results.

3- Page 24. The sentence starting with 'For example, ...' is not a sentence. Please revise.

4- Figure 2. Consider adding means and confidence intervals.

5- Figure 3. Please include a legend to show which traces are with and without vision.

References:

Dalton, B. H., Rasman, B. G., Inglis, J. T., & Blouin, J. S. (2017). The internal representation of head orientation differs for conscious perception and balance control. *The Journal of Physiology*, 595(8), 2731-2749.

Goldberg, J. M., Smith, C. E., & Fernández, C. (1984). Relation between discharge regularity and responses to externally applied galvanic currents in vestibular nerve afferents of the squirrel monkey. *Journal of neurophysiology*, 51(6), 1236-1256.

Kwan, A., Forbes, P. A., Mitchell, D. E., Blouin, J. S., & Cullen, K. E. (2019). Neural substrates, dynamics and thresholds of galvanic vestibular stimulation in the behaving primate. *Nature communications*, 10(1), 1904.

Lund, S., & Broberg, C. (1983). Effects of different head positions on postural sway in man induced by a reproducible vestibular error signal. *Acta Physiologica Scandinavica*, 117(2), 307-309.

Wardman, D. L., Taylor, J. L., & Fitzpatrick, R. C. (2003). Effects of galvanic vestibular stimulation on human posture and perception while standing. *The Journal of physiology*, 551(3), 1033-1042.

END OF COMMENTS

Response to the reviewers' comments for the manuscript:

“Visual modulation of vestibular-evoked balance response disturbed by posterior cortical atrophy”.

Reviewing Editor:

In addition to the reviewers' comments, the authors need to provide a table listing the VBM analysis results - MNI co-ordinates of each cluster, z value, cluster size as well as the mean SEM grey matter density values at a minimum. The authors also need to describe the second level statistical approach in more detail. The figures need to be improved by adding slice locations and cluster location labels.

Thank you for this comment. We have added a table listing the VBM analysis results on page 31. We have also added slice locations and cluster location labels to Figures 4 (VBM) and 5 (TBSS).

We have also expanded the VBM statistical approach in more detail (p12):

“We employed SnPM’s Multisubject Correlation model to examine within-group relationships between modulated grey matter volume and GVS response measures, using a multiple regression design matrix with continuous behavioural regressors.”

Reviewer 1:

The authors report on group differences between healthy controls, Alzheimer and PCA patients around a GVS-induced postural response. Concurrent neuroimaging data was used to interpret the findings. The study is robust and sound, its findings though unsurprising. My addressable concerns are with respect to some methodological details, the framing of the findings and the discussion.

We thank you for your positive comments, including on the robustness of the study and imaging methods (R1, point 9).

We consider two findings to be potentially surprising. Firstly, that the reduced visual modulation of the GVS-evoked balance response in the PCA group is occurring even before the opportunity for visual signals of self-motion (Visuo-Vestibular Integration; R1, point 1). Secondly, that patient group responses to GVS without vision were appropriately timed and directed despite i) extensive brain atrophy following years of neurodegeneration and ii) demands for transforming vestibular signals across reference frames between the head and feet (Proprioceptive-Vestibular Integration; R2, point 1).

1. *Title - In my interpretation of the results from the perspective of a PCA patient to the artificial vestibular stimulus is not an aggravating disturbance of the postural response but rather an attenuation due to a reduced visual motion processing in these patients. I would suggest to rephrase it a bit.*

We agree that reduced visual motion processing is likely a key component of altered visual feedback effects, and had originally noted this in the Discussion. However, we also noted that the reduced modulating effect of vision on GVS-evoked balance responses cannot solely be explained by altered visual feedback (presumably, diminished processing of optic flow caused by GVS-evoked balance response itself).

The smaller attenuating effect of vision in the PCA group was evident during the earliest mechanical indicator of GVS responses detected by force plate measurement between 200-400ms post-stimulus onset. In healthy participants, visual feedback effects have little impact before 400ms post GVS onset. This is a key finding which suggests a feedforward reduction in the weighting of visual information towards vestibular-evoked balance response. While feedforward reductions have been noted following deafferentation in early life, the current results provide evidence of feedforward re-weighting following onset of neurodegenerative disease in the sixth decade of life. It is plausible that the feedforward reduction of vision is itself a consequence of long-term deterioration of visual motion processing owing to PCA. We have now rephrased the Discussion (p18):

“[...] attenuating vestibular-evoked balance response. [...] In the current PCA group, feedforward reduction of visual input from cortex onto presumed subcortical vestibulo-spinal pathways may itself arise from progressive deterioration of visual motion processing, discussed below with reference to neuroanatomical substrates.”

2. *PCA as a disease is not known for gait disturbances or an increased fall risk in the clinical neurology literature. PCA is not known for lateropulsion, Pusher syndrome or loss of trunc uprightness at all. Please give references for this in the introduction or drop the point.*

We had originally included a reference (Crutch et al., 2018) which reports injuries from falls and loss of trunk uprightness shared by PCA patients along with quotes.

We have now introduced additional references in the Introduction from clinical neurology reviews emphasising increased falls risk in PCA (Graff-Radford et al., *Lancet Neurol.*, 2021; Yong et al., *Curr Treat Options Neurol.*, 2023). In addition to Crutch et al. (2018), we introduce a reference on postural abnormalities in late-stage AD (Van Engelen et al., 2020) (p5).

3. *Lesions studies only implicate the lateral thalamus and the superior temporal gyrus in verticality perception and not the parietal lobe (see works by Karnath, Baier and Perennou).*

We agree that altered verticality perception may arise from thalamic and/or superior temporal lesions following strokes. However, this does not preclude parietal contributions to verticality perception. Correspondingly, we note that Perennou et al. concluded from 86 stroke survivors: “Finally, lesions in all central nervous system (CNS) areas could produce dissociations between modalities of verticality perception with the exception of parietal cortex where lesions biased all modalities of verticality perception.

These findings also support the view that the parietal cortex plays a critical role in the multimodal perception of the vertical, which is a condition for elaborating an internal model of verticality.” (Perennou et al., Brain, 2008, p.2412)

Of key relevance to the current work is a recent study of visual and haptic verticality perception in PCA and typical AD (Day et al., JPhysiol, 2022). To our knowledge, this is the only formal investigation of verticality perception in PCA and typical AD to date. Of note: 1) altered verticality perception was documented in these two patient groups with neurodegenerative (not stroke) aetiology, 2) greater disturbances in verticality perception (PCA>tAD) were associated with superior parietal atrophy, and 3) almost all of the participants from Day et al. (2022) underwent current study procedures.

We now expand the Introduction to mention superior temporal and thalamic regions being implicated in verticality perception (p6):

“While superior temporo-parietal and thalamic regions have been implicated as key sites in verticality perception (Perennou et al., 2008), [...]”

4. *Short-duration GVS in primates induces a stimulation of semicircular canals and otolith organs alike inducing a head tilt in the frontal plane (PMID: 31015434). Not a head rotation. Please correct.*

As far as we can see, the work cited by the reviewer does not show a head tilt response to GVS but an ocular torsion. However, we agree that the literature indicates equal activation of all semicircular canal and otolith afferents by GVS, which we now include with citations (see below).

The question then becomes what is the **net** effect of this uniform activation and how do central processes act on it? Fitzpatrick and Day (2004) modelled such an activation and concluded that the otolith signals mutually cancel while the semicircular canal signals sum vectorially to mimic a rotation approximately about the roll axis in head coordinates. This modelled response has received experimental support in human studies involving rotation perception (Day & Fitzpatrick, J Physiol., 2005), gait (Fitzpatrick et al, Current Biol., 2006) and balance (Mian et al, J Physiol., 2010). Therefore, we feel justified in maintaining that GVS creates a virtual rotation of the head in space.

We now include additional references and expand Introduction text for clarification (p6):

“The vestibular perturbation consists of a small electric current delivered to the mastoid processes (galvanic vestibular stimulation; GVS) that acts to modulate the firing rate of semi-circular canal and otolith afferents (Goldberg et al., 1984; Kwan et al., 2019). According to one model, the net effect of this vestibular afferent activity mimics a vestibular signal consistent with an apparent rotation of the head approximately about its roll axis (Fitzpatrick & Day, 2004). This modelled response has received experimental support in human studies involving rotation perception (Day & Fitzpatrick, 2005), gait (Fitzpatrick et al, 2006) and quiet standing (Mian et al, 2010). When standing still with

head upright, this equates to an apparent fall of the body that is rapidly compensated by an opposing motor response to maintain balance (Day et al, 1997).”

5. *The grey matter loss in PCA patients affect the entire dorsal visual stream. The vestibular integration of visual motion information via area 3A, 6 into area CSv for egomotion detection (see Figure 5a of the manuscript (PMID: 21709176 PMID: 36334557) and the intraparietal sulcus (VIP) for grasping (PMID: 15872109). This topic should be addressed also with respect to the previous work in this journal. The interesting findings in PCA also shine a methodological light on how we measure verticality perception visually and haptically and the role of the parietal cortex.*

Thank you for noting the interest in findings and methodology. We now include those references relevant to egomotion detection on page 18:

“Occipito-temporal correlates overlap with neuroanatomical correlates of self-motion detection, sensorimotor transformations necessary for movements such as grasping, visual motion discrimination and visual crowding in PCA, relevant for the fidelity of visual contributions to balance (Orban *et al.*, 1998; Schlack *et al.*, 2005; Fischer *et al.*, 2012; Yong *et al.*, 2014; Ruehl *et al.*, 2022).”

We have included references with respect to previous work in this journal (see R2, point 1).

6. *Table 1 shows a significant difference in the vibrotactile results (the unit is unclear from the caption though) from all plantar testing points reflecting a higher sensitivity to proprioceptive input from the foot. This finding is not sufficiently addressed and discussed in the manuscript. Better proprioception from the feet to counterbalance loss of sensitivity in visual motion processing?*

Vibrotactile sensitivity was comparable (mid-tibia) or increased (plantar) in PCA relative to Control and tAD groups based on median vibrotactile thresholds (μm). We agree this is a potentially interesting finding which may warrant further investigation. However, we note that the PCA group is younger than the other groups and had reported age-adjusted results correspondingly (Results, p16).

We now introduce the following text in the Conclusions (p20):

“While younger age may account for lower observed vibrotactile thresholds on plantar sites in the PCA group, further work might evaluate somatosensory balance control (e.g. effect of light touch with/without vision) to investigate the possibility of altered visual-somatosensory integration.”

“Table 1 shows a significant difference in the vibrotactile results (the unit is unclear from the caption though)”

Thank you for noting the omitted units - we now include these in Table 1. We had not reported statistical tests for Table 1 in line with STROBE guidelines on avoiding significance tests in descriptive tables.

7. *None of the neuropsychological tests include testing for visual motion sensitivity and specificity. Please address.*

Thank you for this comment. We now note this limitation in the manuscript (p20):

“Fourth, the absence of neuropsychological measures specifically assessing visual motion sensitivity or specificity. Future studies aiming to characterise the role of visual motion processing in PCA or related conditions would benefit from including targeted tests of optic flow perception, motion coherence, or direction discrimination.”

8. *GVS stimulus: Length of rectangular stimulus unclear. Why does the electrical current onset and its offset not represent two separate events that should be modelled (PMID: 36732070)?*

We had previously stated the duration of vestibular stimulation in Methods. We now additionally report this in the Figure 1 legend: “Trial timing (2s baseline, 2s post-GVS onset, 2s post-GVS offset) [...]”.

The reviewer is correct to point out the responses to current onset and offset represent two separate events. The response to current offset is approximately opposite to the onset response, as is evident from the force record in Fig 1. However, the offset occurs while the participant is actively responding to the maintained current rather than from a quiescent state. Essentially, while all participants are standing upright at GVS onset, each participant is much more variable in their active response at GVS offset. This raises a potential interpretation problem because the body state is uncontrolled when the stimulus is turned off and most likely varies considerably between participants. For this reason, we choose not to quantify the off-response.

9. *Neuroimaging analysis: Great application of permutation testing via SnPM to produce robust findings in small cohorts.*

Thank you for noting the robust findings.

10. *Why not include the white matter findings from the VBM analysis to complement and support the TBSS analysis?*

We thank the reviewer for this thoughtful suggestion. We chose not to perform a white matter VBM analysis because this method is less suited to detecting white matter differences for several methodological reasons. First, T1-weighted images have lower contrast for white matter than for grey matter, increasing the risk of tissue misclassification and noise. Additionally, the high variability in white matter tract orientation across individuals poses challenges for VBM’s registration algorithms, which may result in suboptimal alignment and blurred group effects. Spatial smoothing,

commonly used in VBM to manage variability, is particularly problematic for white matter, as it can exacerbate partial volume effects and obscure subtle differences. Finally, VBM is primarily sensitive to macroscopic volumetric changes, whereas many clinically relevant white matter alterations occur at the microstructural level and are better captured by diffusion-based methods such as TBSS. For these reasons, we considered TBSS to be the more appropriate approach for our study aims.

11. *Throughout the neuroimaging findings the anatomical localization reporting for the grey matter and the fiber tracts appear a little outdated (see <https://www.ebrains.eu/tools/human-brain-atlas>) and might be improved.*

Thank you for the valuable suggestion regarding the use of updated anatomical atlases. In our study, we used the Harvard-Oxford cortical and subcortical structural atlases and JHU ICBM-DTI-81 white matter labels and tracts atlases. These atlases were selected because they are derived from large, in vivo datasets validated against extensive normative samples, providing high reliability for group-based comparisons. Additionally, their widespread adoption in the neuroimaging community facilitates comparability across studies. While the EBRAINS atlas provides high-resolution postmortem data, its basis in more limited datasets makes it less suited for representing typical population anatomy in our study context. For these reasons, we retained the current atlas choices to ensure methodological consistency and reliability.

We now include additional text to Methods for clarification (p13):

“We used the Harvard-Oxford cortical and subcortical structural atlases and JHU ICBM-DTI-81 white matter labels and tracts atlases, validated in vivo datasets and widely adopted in neuroimaging research, to label significant grey matter regions and white matter tracts, ensuring both reliability and comparability across studies.”

Reviewer 2:

Overall, the paper is well written but there are a number of points the authors should consider.

Thank you for your comments and points which we address below.

1. *The authors mentioned in the first paragraph of the Introduction the presence of a 'coherent perception of body orientation and movement in space' to justify their study. However, it is not clear why the authors developed these arguments as there are numerous examples in the literature where perception and action don't line up - some even specific to the vestibular control of balance (see Dalton et al. 2017 & Wardman et al. 2003). Consequently, the authors should present a balanced perspective in the Introduction to justify their experiment and hypotheses. Also, their results should be interpreted in line with the known distinctions between balance actions and perception in the Discussion.*

Our original Introduction emphasised several functions of key relevance to both perception of upright and balancing the upright body: visual processing, multisensory integration and spatial transformations. Current study hypotheses are essentially predicated on disruption of these functions - spatial transformations and visual processing - in the context of posterior parietal and occipito-temporal atrophy.

We agree that perception and action dissociate, including in the perception of head orientation and vestibular balance control. Indeed, we consider perceptual and balance functions to be separable throughout the Discussion and Conclusions.

However, the key dissociation of interest is not between perception and action, but between the spatial transformations required for visual and haptic vertical perception (e.g. head- to eye- and body-centred frames) and those required for appropriately directing balance response based on orientation of the head relative to the feet (head-to body- and effector frames). There is evidence that the former have been disrupted in both patient groups (Day et al., JPhysiol, 2022; R1, point 3.). In contrast, the latter appear comparable across control and patient groups based on GVS responses without vision (Proprioceptive-vestibular integration). This suggests separability of systems responsible for spatially transforming these sensory signals.

We now expand Introduction and Conclusion sections:

“Consequently, disorders that affect parietal regions and associated functions may therefore compromise the ability to stabilise the body while standing.” (p5)

“Furthermore, we consider current findings in light of disturbances in verticality perception to better understand well documented dissociations between perceptual and sensorimotor functions relevant to vestibular control of balance (Dalton et al. 2017 & Wardman et al. 2003).” (p6)

“Discrepancies between perceptual and balance functions align with prior work demonstrating divergence between perceived orientation and vestibular-evoked balance responses (Wardman et al., 2003; Dalton et al., 2017).” (p20)

2. *Interpretation of visual modulation. The authors observed larger GVS-evoked responses in participants with posterior cortical atrophy when vision was present. These results may indicate a higher gain of the vestibular channel in this group of participants. The authors, however, interpret these results as an indication of a reduction in the weighting of visual inputs onto vestibulo-spinal pathways. To accept this argument, one would have to present evidence that the gain of other sensory channels did not change in these participants, or support this statement with a computational model of the system.*

Higher gain of the vestibular channel would result in abnormally large GVS-evoked responses, particularly without vision. Previous work in a patient with complete loss of somatosensation (patient IW) has identified markedly elevated GVS-evoked responses, reaching ≥ 10 times the magnitude of displacement response compared to controls without vision at similar GVS amplitude (Day & Cole, Brain, 2002).

In the absence of vision, GVS response magnitude was comparable between PCA and control groups based on measures of force (controls: 6.26N [4.72, 7.80]; PCA: 6.33N [4.64, 8.02]) or displacement (controls: 13.69mm [11.05, 16.34]; PCA: 15.90mm [12.58, 19.23]) (Figure 2; Figure 3). We therefore consider the smaller effect of vision on GVS-evoked responses in the PCA group to be consistent with reducing weighting of visual input in attenuating vestibular-evoked responses.

3. *Related to the first major comment, the authors should consider citing the proper literature throughout the text. For example, Fitzpatrick & Day (2004) did not record vestibular afferents (page 5) but proposed a model to understand the effects of GVS. Please cite the original physiological papers (see Goldberg et al. 1984 & Kwan et al. 2019). Similarly, cite the original papers related to the craniocentricity of vestibular responses (Lund & Broberg, 1983).*

We agree on the citing the original physiological papers- the original citation (Fitzpatrick & Day, J Appl Physiol, 2004) was an invited review which cited the suggested papers (Goldberg et al., 1984; Lund & Broberg, 1983).

We now cite these papers along with Kwan et al (2019) in the Introduction (p6).

“The vestibular perturbation consists of a small electric current delivered to the mastoid processes (galvanic vestibular stimulation; GVS) that acts to modulate the firing rate of semi-circular canal and otolith afferents (Goldberg et al., 1984; Kwan et al., 2019). According to one model, the net effect of this vestibular afferent activity mimics a vestibular signal consistent with an apparent rotation of the head approximately about its roll axis (Fitzpatrick & Day, 2004). This modelled response has received experimental support in human studies involving rotation perception (Day & Fitzpatrick, 2005), gait (Fitzpatrick et al, 2006) and quiet standing (Mian et al, 2010). When standing still with head upright, this equates to an apparent fall of the body that is rapidly compensated by an opposing motor response to maintain balance (Day et al, 1997).”

We also cite original physiological papers related to the effect of vision on GVS response magnitude in the Discussion (p18): “(Smetanin et al., 1990; Day & Bonato, 1995; Day & Guerraz, 2007).”

4. *Where were the CODA markers located on the pelvis?*

We originally reported this under Methods. We have now expanded the text (p8):

“[...] shoulders (left and right acromion process), pelvis (left and right superior iliac crest), [...]”

5. *Were conditions randomized between participants or were the conditions performed in the order presented in the paper (page 9)? Please justify if there was no randomization.*

All conditions were randomized. We now rephrase for clarity (p9):

“All conditions were randomized across trials.”

6. *The GVS amplitude was 1 mA and participants were exposed to 10 trials per condition/polarity combination. Please show the variability in the responses within and between participants given the low number of trials and small GVS magnitude may lead to variation in responses, particularly in the groups tested. The authors need to convince the reader that the evoked force and movement responses had high signal to noise ratios.*

We originally presented measures of within participant variability (angular deviation in force and displacement response (deg); previous Figure 2A). Intraindividual variability measures were comparable between participant groups. Furthermore, estimated 95% confidence intervals were narrow for these and other evoked force and displacement responses.

We show response magnitude and within-participant variability below on the same log scale (Figure). These are consistent with high signal to noise ratios; averaging across groups and conditions, intraindividual variability measures were approximately 57% to 50% of evoked force or neck-level displacement responses respectively (estimated geometric mean force: 4.6N; intraindividual SD: 2.6N; displacement: 10.5mm; intraindividual SD: 5.3mm).

We note that a key finding- abnormally elevated PCA response magnitude with vision - was accompanied by proportionately elevated PCA intraindividual SD. However, we limit the below Figure to our response given intraindividual SD inherently includes a measure of response magnitude.

Additionally, we now show 95% CIs in Figure 2 as requested (R2, point 21.).

Figure. Effect of vision on GVS response magnitude (left) and variability (intraindividual standard deviation, right) in control, tAD and PCA groups. Response magnitude and variability is shown without and with vision with the head straight ahead. Responses are determined by force plate (top) and neck-level displacement (bottom). Participant-level observed mean response magnitude and SD are presented with estimated marginal means and 95% CIs on a log scale.

7. P10. The authors should justify why they used constant time points to determine the GVS-evoked responses.

Thank you for this comment. The selected time windows are based on previous studies examining the temporal dynamics of GVS-evoked postural responses. Specifically, the 200–400 ms window captures the initial peak force response at the feet, which reflects early compensatory adjustments during the destabilising phase before whole-body movement occurs. The 300–800 ms window corresponds to the later phase, during which whole-body postural adjustments emerge.

We have clarified this rationale in the manuscript and added relevant references (p9):

“Time windows for analysis were based on previous studies characterising the temporal profile of GVS-evoked postural responses (e.g., Day & Guerraz, 2007; Fitzpatrick et al., 1994; Wardman et al., 2003). The 200–400 ms window captures the initial force response at the feet, representing early compensatory postural activity following the onset of vestibular perturbation. The 300–800 ms window reflects the later, more global body sway response, which takes longer to develop and involves whole-body postural adjustments incorporating sensory feedback”

8. Show your equations for determining the direction and angular deviation of the GVS-evoked responses...

We now include equations for direction, magnitude and angular deviation in Methods (p10):

“Each participant's mean response measures were calculated as:

$$\text{Mean response magnitude} = \frac{1}{n} \sum_{i=1}^n \sqrt{x_i^2 + y_i^2}$$

$$\text{Mean response direction} = \arctan \left[\left(\frac{\left(\frac{1}{n} \sum_{i=1}^n (\sin \theta_i) \right)}{\left(\frac{1}{n} \sum_{i=1}^n (\cos \theta_i) \right)} \right) \right]$$

$$\text{Angular deviation} = \sqrt{2(1 - r)}$$

Where:

n = number of trials

x_1 = mediolateral component of the response on the i -th trial

y_1 = anteroposterior component of the response on the i -th trial

θ_i = response direction on the i -th trial relative to the interaural line, indicating a response directed perfectly towards the anodal ear and positive values indicating responses counterclockwise of the anodal ear (Figure 1C)

$$r = \frac{1}{n} \sqrt{(\sum_{i=1}^n \cos \theta_i)^2 + (\sum_{i=1}^n \sin \theta_i)^2}$$

...The authors mentioned a 0 deg reference system for anodal but this is not clear as it would lead to a head fixed coordinate system. Please show your reference system in Fig 1 and include both the AP and ML responses so the reader can visualize how the angular values are computed...

We used a head fixed coordinate system in which responses are determined relative to the anodal ear, irrespective of the orientation of the head. An advantage of this system is that this minimises the range of deviation from zero and limits the need for circular statistical methods (see R2 point 10).

We further expand Figure 1 and the legend to show this reference system and include AP and ML responses.

...Also represent the vector of the evoked responses in Fig 2 to show both the response magnitude and direction for all participants and conditions.

We now show vectors in Figure 2 showing response magnitude and raw direction. For consistency with Figure 2 measures and analyses, we show these for polarity conditions and have swapped left and right anode positions.

9. *Why did the authors average across polarities for the vision comparison but not for the craniocentric experiment?*

In the craniocentric experiment (“Proprioceptive-Vestibular Integration: GVS Responses Across Head Positions”), our primary interest was in assessing response variability, direction and magnitude across head positions in the absence of vision. This required preserving polarity-specific responses, as polarity determines the direction of the evoked response. The effect of polarity on balance responses was comparable across groups.

The vision experiment (“Visuo-Vestibular Integration: Effect of Vision on GVS Response”) compared response magnitude with versus without vision with the head straight ahead. We evaluated effects of vision on response magnitude irrespective of polarity for the following reasons. Firstly, we did not have an a priori reason that the effect of vision on the direction of the GVS-evoked response would differ between groups. We did however hypothesize that visual processing deficits in PCA would be associated with reduced visual modulation of balance response magnitude (R2, point 3). Secondly, we sought to limit unnecessary comparisons (see R2, point 12 and point 16).

10. *Statistics. Please justify why the authors did not use circular statistics. How did they deal with the 0-360 deg (or -180-180 deg) discontinuities in the angular data*

We calculated mean direction and angular deviation using circular summary statistics. However, we conducted formal statistical analysis on responses using a head fixed coordinate system (see point 8, above).

The head fixed coordinate system has the advantage of limiting the range of responses, with participant force and displacement responses ranging from -27.26-39.28 and -53.34-30.88 respectively (Figure 2B). This limits the need for circular statistical methods to analyse data ranging from 0-360/-180-180 deg.

Furthermore, we checked model assumptions regarding normality and took steps to better meet these assumptions (see next point 11).

11. *Statistics. Explicitly state what drove the non-normality in the data. This is particularly important given the main result is related to the vision-no-vision comparison.*

Non-normality was driven by rightwards, or positive, skew. This rightwards skew was evident across groups and is unsurprising given that response magnitude measures are all bounded at 0 but lack a common upper bound.

We used a log transformation to better meet model assumptions regarding normality. Residual plots confirmed that the assumption of normality of residuals was better met

when using log-transformed responses. We estimated effects using log-transformed responses and back transformed estimates to geometric means to aid interpretability.

12. *Statistics. The authors mentioned interactions were included if there 'was evidence of improved model fit'. Please justify why the interaction terms were not always included in the statistical model.*

Interaction terms were included in the statistical models only when they provided a statistically meaningful improvement in model fit, as assessed by likelihood ratio tests. This approach was adopted particularly to limit multiple testing and avoid overfitting.

We have now clarified this rationale in the Methods section (p12):

“To limit multiple testing, interaction terms were included only if there was a priori justification and/or evidence of improved model fit based on a likelihood ratio test.”

13. *Why did the authors change the number of permutations (5000 vs 30000) between the neuroimaging tests?*

The difference in number of permutations arises largely from the distinct nature of the analyses and their demands for statistical precision. In the case of TBSS, the analysis is performed on a skeletonized representation of the tracts, which significantly reduces the number of voxels (and thus tests). This reduction allows 5000 permutations to be sufficient for accurately estimating the null distribution while maintaining a balance between statistical power and computational efficiency. By contrast, whole-brain VBM analyses involve a much larger number of comparisons. Consequently, a greater number of permutations (in our case, 30,000) is necessary to achieve a more precise approximation of the null distribution, despite the associated increase in computational cost. This higher permutation count represents a practical compromise between enhancing the accuracy of our statistical estimates and keeping processing times within acceptable limits.

We have now clarified this rationale in the Methods section (p13):

“The lower permutation number for TBSS reflects the reduced number of comparisons due to skeletonization, allowing accurate estimation of the null distribution with fewer permutations. In contrast, the higher number for VBM was chosen to ensure sufficient precision given the greater voxel-wise testing burden.”

14. *Angular direction of the GVS evoked responses. The authors reported values between -0.27 and -0.93 deg for the anode right and anode left conditions (in text) and Figure 2 shows a near zero angle across conditions. Shouldn't the values between anode right and left show a near 180 deg difference? This point goes back to the Methods points above.*

We calculated responses relative to the anodal ear. See above points 8 and 10 on head-referenced errors and corresponding amendments to Figure 1 and Methods (p10).

15. *Results, page 17. Please describe how the interaction terms were decomposed and clearly show the results for the interaction.*

We originally reported p values corresponding to pairwise comparisons of the effect of vision between groups in the Results (PCA vs controls; PCA vs tAD; tAD vs controls). These correspond to tests for interactions (rather than tests for group differences at each vision condition). To clearly show the interaction, we also reported the comparison of the effect of vision within- and between- groups expressed as ratios in Table 3.

To show results comparing the effect of vision across all groups, we now additionally report the global test along with pairwise differences in the effect of vision in Results (p15 and p16):

“Formal tests of differences between interaction terms provided evidence that the effect of vision in modulating force plate responses was smaller in PCA relative to both control and tAD groups (PCA vs controls: $p=0.002$; PCA vs tAD: $p<0.001$; global $p<.001$).”

“Formal tests of differences between interaction terms provided evidence that the effect of vision in modulating neck-level displacement responses was smaller in PCA relative to both control and tAD groups (PCA vs controls: $p=0.003$; PCA vs tAD: $p<0.001$; global $p<.001$).”

“Formal tests of differences between interaction terms provided evidence that the effect of vision in reducing neck-level displacement responses was smaller in PCA relative to both control and tAD groups (PCA vs controls: $p=0.003$; PCA vs tAD: $p<0.001$; global $p<.001$).”

16. *Results, correlation and neuroimaging. The authors should clearly justify why they performed the correlations with both the force and position signals. What are the specific hypotheses or physiological mechanisms tested? Also, clearly justify why were the correlation analyses only performed within the groups of participants”*

We performed neuroimaging correlation analyses of force and displacement responses separately. While these responses are related, they capture different complementary aspects of balance responses. Force responses reflect early mechanical output at the level of the feet before the opportunity for visual feedback, while displacement responses capture later whole-body adjustments incorporating sensory feedback (see R2 point 7).

Neuroimaging correlation analyses were conducted within groups (rather than across all participants) to avoid confounding by group-level differences in both behaviour and brain structure.

We have clarified these points in the Methods section (p13).

“Correlation analyses were conducted separately within each group to avoid confounding neuroimaging analysis of GVS balance responses with clinico-radiological group differences.”

17. *Page 7. One sentence (starting with 'Conversely,...') is mostly repeated twice.*

Thank you for pointing out the repetition. The repeated sentence has been removed and adjusted (p8):

“The tAD group showed predominant episodic memory and language impairments, with a subset also presenting corticovisual difficulties (Table 2).”

18. *Page 7. Consider presenting the main highlights or the tests performed and the associated results.*

We have restricted tests to those most relevant for establishing PCA criteria.

19. *Page 24. The sentence starting with 'For example, ...' is not a sentence. Please revise.*

Thank you for this helpful comment. We have revised the sentence, replacing “for example” with “[...] such as [...]” (p20)

20. *Figure 2. Consider adding means and confidence intervals.*

We now add estimated means and 95% CIs for each group. We have removed these from the Results text to limit repetition.

21. *Figure 3. Please include a legend to show which traces are with and without vision.*

We have now adjusted traces (without vision: full line; with vision: dashed line) and added a legend and expanded legend text (p32):

“[...] Full and dashed lines show responses without and with vision, darker and lighter colours show responses with anode right and left respectively. B) Response magnitude determined by force plate (top) and neck-level displacement (bottom), averaged across polarity conditions. Mean observed response magnitude is presented with estimated marginal means and 95% CIs overlaid in black. “

Dear Dr Yong,

Re: JP-RP-2025-288693R1 "Visual modulation of vestibular-evoked balance response disturbed by posterior cortical atrophy" by Dilek Ocal, Brian L Day, Amy Lynne Peters, Matthew Bancroft, David Cash, Diego Kaski, Natalie Ryan, Sebastian Crutch, and Keir Yong

Thank you for submitting your manuscript to The Journal of Physiology. It has been assessed by a Reviewing Editor and by 2 expert referees and we are pleased to tell you that it is acceptable for publication following satisfactory revision.

REVISION CHECKLIST:

Please upload two versions of your manuscript text: one with all relevant changes highlighted and one clean version with no changes tracked. The manuscript file should include all tables and figure legends, but each figure/graph should be uploaded as separate, high-resolution files. The journal is now integrated with Wiley's Image Checking service. For further details, see: <https://www.wiley.com/en-us/network/publishing/research-publishing/trending-stories/upholding-image-integrity-wileys->

image-screening-service

We look forward to receiving your revised submission.

Yours sincerely,

Vaughan Macefield
Senior Editor
The Journal of Physiology

EDITOR COMMENTS

Reviewing Editor:

Dear authors,

Thank you for addressing the reviewers comments. There remain 2 very minor changes suggested:

Line 175: Consider changing stance to head orientation or head posture.

Should subtitle A) in Table 1) be Demographic information instead of Main effect of vision?

Senior Editor:

Thank you for submitting your revised manuscript to The Journal of Physiology. I am pleased to report that your manuscript will be acceptable for publication once you attend to the minor comments raised by Reviewer 2.

REFEREE COMMENTS

Referee #1:

The authors have addressed all of my concerns.

Referee #2:

The authors addressed all comments from the initial review. Please find very minor suggestions for the authors to consider:

1- Line 175: Consider changing stance to head orientation or head posture

2- Should subtitle A) in Table 1) be Demographic information instead of Main effect of vision?

END OF COMMENTS

Response to the reviewers' comments for the manuscript:

"Visual modulation of vestibular-evoked balance response disturbed by posterior cortical atrophy".

Reviewing Editor:

Dear authors,

Thank you for addressing the reviewers comments.

Thank you, the Senior Editor and reviewers for their review and comments.

There remain 2 very minor changes suggested:

Line 175: Consider changing stance to head orientation or head posture.

We have now changed 'stance' to 'head orientation' for consistency.

Should subtitle A) in Table 1) be Demographic information instead of Main effect of vision?

We have now amended subtitle A) in Table 1 to 'Demographic information'.

Dear Dr Yong,

Re: JP-RP-2025-288693R2 "Visual modulation of vestibular-evoked balance response disturbed by posterior cortical atrophy" by Dilek Ocal, Brian L Day, Amy Lynne Peters, Matthew Bancroft, David M Cash, Diego Kaski, Natalie Ryan, Sebastian Crutch, and Keir Yong

We are pleased to tell you that your paper has been accepted for publication in The Journal of Physiology.

Yours sincerely,

Vaughan Macefield
Senior Editor
The Journal of Physiology

If you would like to receive our 'Research Roundup', a monthly newsletter highlighting the cutting-edge research published in The Physiological Society's family of journals (The Journal of Physiology, Experimental Physiology, Physiological Reports, The Journal of Nutritional Physiology and The Journal of Precision Medicine: Health and Disease), please click this link, fill in your name and email address and select 'Research Roundup':
<https://www.physoc.org/journals-and-media/membernews>

- You can help your research get the attention it deserves! Check out Wiley's free Promotion Guide for best-practice recommendations for promoting your work at: www.wileyauthors.com/eeo/guide. You can learn more about Wiley Editing Services which offers professional video, design, and writing services to create shareable video abstracts, infographics, conference posters, lay summaries, and research news stories for your research at: www.wileyauthors.com/eeo/promotion.

EDITOR COMMENTS

Senior Editor:

Thank you for attending to these minor comments. I am pleased to report that your manuscript is now considered acceptable for publication in The Journal of Physiology.